# Identification of circulating proteins associated with general cognitive function among middle-aged and older adults

Adrienne Tin [1,2,47✉], Alison E. Fohner [3,4,5,47✉], Qiong Yang [6], Jennifer A. Brody[5], Gail Davies [7], Jie Yao[8], Dan Liu[9], Ilana Caro [10], Joni V. Lindbohm[11,12,13], Michael R. Duggan [14], Osorio Meirelles[15], Sarah E. Harris [7], Valborg Gudmundsdottir[16,17], Adele M. Taylor[7], Albert Henry[18], Alexa S. Beiser[6,19], Ali Shojaie[20], Annabell Coors [9], Annette L. Fitzpatrick[3,21], Claudia Langenberg [22,23,24], Claudia L. Satizabal[19,25,26], Colleen M. Sitlani[5], Eleanor Wheeler[23], Elliot M. Tucker-Drob [27], Jan Bressler [28], Josef Coresh [29], Joshua C. Bis [5], Julián Candia [30], Lori L. Jennings [31], Maik Pietzner [22,23,24], Mark Lathrop[32], Oscar L. Lopez [33], Paul Redmond[7], Robert E. Gerszten [34], Stephen S. Rich [35], Susan R. Heckbert [3], Thomas R. Austin[3,5], Timothy M. Hughes[36,37], Toshiko Tanaka[30], Valur Emilsson [16,17], Ramachandran S. Vasan[19,38,39], Xiuqing Guo [8], Yineng Zhu[6], Christophe Tzourio[10], Jerome I. Rotter [8], Keenan A. Walker [14], Luigi Ferrucci [30], Mika Kivimäki [40,41], Monique M. B. Breteler [9,42], Simon R. Cox [7], Stephanie Debette [10,43], Thomas H. Mosley[1], Vilmundur G. Gudnason [16], Lenore J. Launer [44], Bruce M. Psaty [3,5,45], Sudha Seshadri [19,25,48] & Myriam Fornage [28,46,48]

Identifying circulating proteins associated with cognitive function may point to biomarkers and molecular process of cognitive impairment. Few studies have investigated the association between circulating proteins and cognitive function. We identify 246 protein measures quantified by the SomaScan assay as associated with cognitive function ($p < 4.9E-5$, n up to 7289). Of these, 45 were replicated using SomaScan data, and three were replicated using Olink data at Bonferroni-corrected significance. Enrichment analysis linked the proteins associated with general cognitive function to cell signaling pathways and synapse architecture. Mendelian randomization analysis implicated higher levels of NECTIN2, a protein mediating viral entry into neuronal cells, with higher Alzheimer's disease (AD) risk ($p = 2.5E-26$). Levels of 14 other protein measures were implicated as consequences of AD susceptibility ($p < 2.0E-4$). Proteins implicated as causes or consequences of AD susceptibility may provide new insight into the potential relationship between immunity and AD susceptibility as well as potential therapeutic targets.

A full list of author affiliations appears at the end of the paper.

Poor cognitive function may be a precursor of dementia, which is expected to affect over 153 million people worldwide by 2050[1]. No treatment exists for curing dementia, and some clinical trials are thought to have failed because the intervention was too late in the pathologic process[2]. Prior large-scale analyses have identified genetic variants associated with general cognitive function, which has significant genetic correlation with Alzheimer's Disease (AD), the most common form of dementia[3]. Compared with genetic variants, protein levels may more closely reflect biological activity. The circulating proteome includes secreted and tissue leakage proteins, which can inform health status and disease risk[4]. Circulating proteins might be a marker of vascular dysfunction[5], which has long been hypothesized as an important component of AD pathophysiology[6]. Identifying plasma proteins associated with general cognitive function may provide meaningful insight into the biological processes related to cognitive function and dementia development.

Prior studies have shown associations of plasma proteins with cognitive differences and incident dementia among middle-aged or older adults[7–9]. These proteomic studies and prior genetic studies have implicated the involvement of inflammation and immune dysregulation in the development of dementia and AD[8–11]. By leveraging high-throughput proteomic data across multiple population-based cohorts, we identified circulating proteins associated with general cognitive function and gene sets enriched for these associated proteins. Using Mendelian randomization (MR) analysis, we further identified the potential causal proteins of general cognitive function and late-onset AD susceptibility. We replicated the cognition-associated proteins in additional cohorts with the same and complementary proteomic technology.

## Results

### Characteristics of the participants in discovery and replication analyses.
A flowchart of the primary analyses is presented in Fig. 1. In the discovery analysis, the total sample size was 7277 from three cohorts. The mean age was 46 in the Framingham Heart Study Third Generation cohort (FHS Gen3) and 75 in the Atherosclerosis Risk in Communities (ARIC) study and Cardiovascular Heart Study (CHS). FHS included White participants only. ARIC and CHS included Black and White participants (17%

and 14% Black, respectively, Supplementary Data 1). The protein measures were quantified using an aptamer-based platform (SomaScan, Supplementary Data 2). For the primary replication using the SomaScan platform, the total sample sizes were 8891 for general cognitive function aged ≥25, 5268 for aged ≥65, and 5478 for performance on the Digit Symbol Substitution Test (DSST)[12] aged ≥65 from 4 cohorts (Supplementary Data 3). We also attempted exploratory replication using data from the Olink proteomic platform from 5 cohorts with a total sample size of 2925 for general cognitive function aged ≥25 and 2225 for aged ≥65 and 1744 for DSST aged ≥65, Supplementary Data 4).

### Discovery and replication results.
We use protein measures to refer to the quantified protein levels given that some proteins were quantified by more than reagent. A total of 1049 protein measures annotated to 1043 unique proteins were tested for general cognitive function among participants aged ≥25. For general cognitive function, the first unrotated principal component generated from cognitive scores from 3 or more different domains (Supplementary Data 5–7), we identified 79 significant protein measures (p-value < 4.77E-5 = 0.05/1049, Table 1, Fig. 2 and Supplementary Data 8, 9). Of these, most (n = 70) were also significant for general cognitive function among participants aged ≥65 (Supplementary Fig. 1). In the analysis of 4709 protein measures (annotated to 4506 proteins) from the two discovery cohorts with participants aged ≥65, we identified 211 significant protein measures associated with general cognitive function and 188 with DSST (Supplementary Data 8–10), respectively (p-value < 1.06E-5 = 0.05/4709). Among the significant proteins, we observed high correlation of betas between the two cohorts with participants aged ≥65 (ARIC and CHS, correlation from 0.86 to 0.88) and much lower correlations between these two cohorts and the cohort with largely middle-aged participants (correlation ≤ 0.06, Supplementary Data 11). Across all three meta-analyses, 246 non-overlapping protein measures were significantly associated with at least one of the separate outcomes (Supplementary Fig. 1).

Of the 220 protein measures that were significantly associated with general cognitive function in the discovery analyses among participants aged ≥25 or 65, 20 were associated with incident dementia in the ARIC study, including growth differentiation factor 15 (GDF15), sushi, von Willebrand factor type A, EGF and

**Discovery analysis of the association between circulating proteins and cognitive function**
- **General cognitive function and DSST, n=7,277 from ARIC, CHS, FHS**
- **Protein assay platform: SomaScan, a modified aptamer-based assay**
- **# of protein measures tested: general cognitive function aged ≥ 25 (1,049), general cognitive function and DSST aged ≥ 65 (4,709)**

**Replication**
- **SomaScan, n=8,891 (AGES, BLSA, MESA, Whitehall II)**
- **Olink, n=2,925 (CARDIA, LBC1921, LBC1936, Rhineland, Three City)**

**Enrichment analysis**
- **Input: significant proteins from discovery stage**
- **Background: encoding genes of the proteins tested**

**Two-sample MR analysis**
- **Exposure and outcome**
  - **General cognitive function and Alzheimer's disease**
  - **Protein measures**

**Fig. 1 Flowchart of main analyses.** General cognitive function was represented by the first unrotated principal component of cognitive scores from 3 or more domains. Abbreviation DSST digit symbol substitution test, ARIC Atherosclerosis Risk in Communities, CHS Cardiovascular Heart Study, FHS Framingham Heart Study, AGES age, gene/environment susceptibility – Reykjavik, BLSA Baltimore Longitudinal Study of Aging, MESA multi-ethnic Study of Atherosclerosis, CARDIA Coronary Artery Risk Development in Young Adults, LBC Lothian Birth Cohort, MR Mendelian randomization.

**Table 1 Summary of the associations between circulating proteins and cognitive function.**

| Analysis | | Discovery analysis | | | | | | | | Replication | | | | |
| --- | --- | --- | --- | --- | --- | --- | --- | --- | --- | --- | --- | --- | --- | --- |
| Cognitive score | Age group | # of protein measures tested* | # of unique proteins tested* | Significant protein measures | Significant unique proteins | I2 median (25th, 75th percentile, max) | P (heterogeneity) < 0.05, n (%) | Protein measures for replication, i.e. beta in the same direction, n (%) | # of protein measures at Bonferroni-corrected significance, n (%) | # of unique proteins at Bonferroni-corrected significance, n (%) | # of protein measures at FDR < 0.05, n | # of unique proteins at FDR < 0.05, n |
| General cognitive function | ≥25 | 1049 | 1043 | 79 | 79 | 77 (60, 85) | 57 (74.0) | 38 (48.1) | 11 (28.9) | 11 | 25 (65.8) | 25 |
| General cognitive function | ≥65 | 4709 | 4506 | 211 | 207 | 66 (0, 84.4) | 93 (44.1) | 208 (98.6) | 26 (12.5) | 26 | 83 (39.9) | 82 |
| DSST | ≥65 | 4709 | 4506 | 188 | 184 | 37.9 (0, 77.7) | 54 (28.7) | 185 (98.4) | 31 (16.8) | 30 | 99 (63.5) | 96 |

All results in this table were from proteins assayed using the SomaScan platform.
Discovery studies with participants aged ≥25 (ARIC, CHS, FHS, n total = 7289), aged ≥65 (ARIC, CHS, n total = 6583).
Replication cohorts using the SomaScan platform included 4 cohorts (n = 8891 for aged ≥25 and n = 4789 for aged ≥65).
*Based on the number of proteins selected for replication and also available in the replication cohorts, the Bonferroni-corrected significance thresholds were: general cognitive function aged ≥25: 1.32E-03 = 0.05/38, and aged ≥65: 2.40E-04 = 0.05/208; DSST aged ≥65: 2.70E-04 = 0.05/185.
DSST digit symbol substitution test, FDR false discovery rate.

pentraxin domain containing 1 (SVEP1), and natriuretic peptide B (NPPB)[7, 9], 67 were associated with cognitive decline[7], and 21 were associated with general cognitive ability (Supplementary Data 12)[13]. In addition, in the promoter region of the encoding genes of 33 significant proteins, there were genetic variants associated with general cognitive function at genome-wide significance (Supplementary Data 13). The Spearman correlation between measures from the SomaScan and Olink platforms were available for 99 of the 246 proteins[14]. All correlations were positive with a median of 0.61 (25th, 75th percentile: 0.44, 0.72, Supplementary Data 14).

For replication, we selected protein measures that were significant in the discovery meta-analyses and had effect estimates in the same direction among all cohorts in each discovery meta-analysis (38 for general cognitive function aged ≥25 and 208 for aged ≥65, and 185 for DSST, Table 1). Of these, the numbers of protein measures that were replicated in independent cohorts using data from SomaScan, our primary replication platform, were 11 for general cognitive function aged ≥25 and 26 for aged ≥65, and 31 for DSST aged ≥65 at Bonferroni-corrected significance (Supplementary Data 8, 15 to 17). Post hoc power calculation showed that given the replication sample size available on the SomaScan platform and Bonferroni-corrected significance level, the powers for replicating the median effect size of the discovery meta-analyses were 0.91 for general cognitive function among participants aged ≥25, 0.67 for aged ≥65, and 0.61 for DSST among participants aged ≥65 (Supplementary Fig. 2). Given the limited power for replication particularly among participants aged ≥65, we also report replication at false discovery rate (FDR) < 0.05 —the number of proteins replicated were 25 for general cognitive function aged ≥25 and 83 for aged ≥25, and 99 for DSST aged ≥65. Across the three analyses, the replication rates were 12.5% to 28.9% based on Bonferroni-corrected threshold and 39.9% to 65.8% based on FDR < 0.05 (Table 1, Fig. 2 and Supplementary Data 15 to 17).

Olink data for exploratory replication were available for 31 of the 38 proteins for general cognitive function among aged ≥25 and 133 of the 208 proteins for aged ≥65, and 10 of the 185 proteins for DSST among aged ≥65. One protein ephrin-A4 (EFNA4) replicated in the meta-analyses of general cognitive function for both aged ≥25 and aged ≥65 at Bonferroni-corrected significance (Supplementary Data 18). CUB domain-containing protein 1 (CDCP1) replicated in both general cognitive function aged ≥65 and DSST, and Macrophage scavenger receptor types I and II (MSR1) replicated at Bonferroni-corrected significance in DSST and FDR < 0.05 in general cognitive function aged ≥65. Lithostathine-a-alpha (REG1A) replicated in general cognitive function aged ≥65 at FDR < 0.05. Among these 4 replicated proteins using the Olink platform, the Spearman correlations between SomaScan and Olink measures were ≥0.7 for 3 with EFNA4 having a correlation of 0.2. It is known that multiple factors could affect the correlation of protein measures quantified using the two platforms, including binding affinity of the reagent (aptamer vs. antibody), glycosylation of the protein, and limit of detection of the assay[14]. Given the replication sample size of the Olink platform, the post-hoc powers for replicating the median effect size of the discovery meta-analysis were 0.18 to 0.29 across the 3 analyses.

**Correlations and potential protein–protein interaction between proteins replicated at Bonferroni-corrected significance.** Moderate Pearson correlations were observed between pairs of the proteins that were replicated at Bonferroni-corrected significance using SomaScan data across the three analyses (general cognitive function ≥25 years, general cognitive function

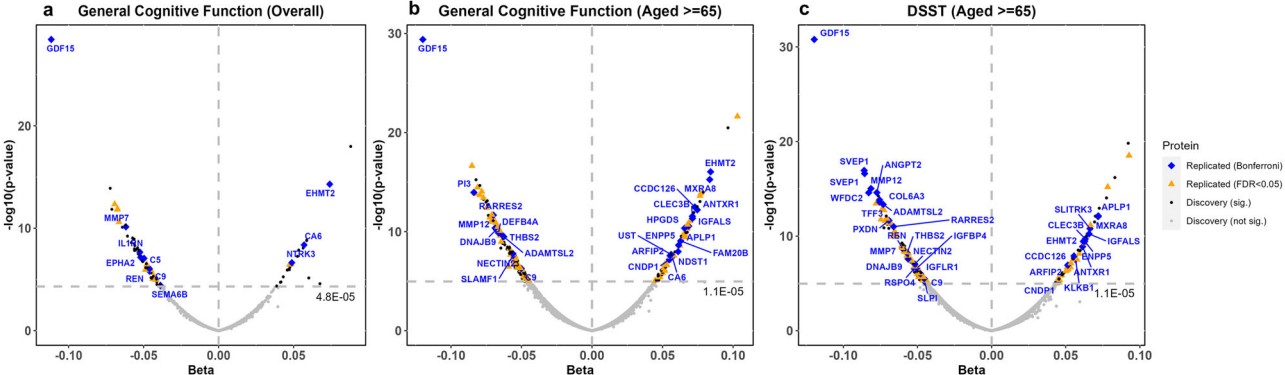

**Fig. 2 Volcano plots showing the beta coefficients and *p*-values from the discovery meta-analyses with colors indicating whether a protein was replicated.** The three discovery analyses were for general cognitive function among aged ≥25 (**a**) and aged ≥65 (**b**), and performance on the Digit Symbol Substitution Test (**c**).

≥65 years, and DSST) with a median absolute correlation of 0.13 to 0.17 (Supplementary Figs. 3–5). Based on protein interaction data from STRING[15,16], proteins that interact with each other are: between C5 and C9 (Supplementary Fig. 6), between CLEC3B and RARRES2 (Supplementary Figs. 7, 8), between COL6A3 and THBS2, and between MMP7 and MMP12 (Supplementary Fig. 8).

**Results of sensitivity analysis controlling for kidney function**. In the sensitivity analysis among the discovery cohorts with participants aged ≥65, after controlling for estimated glomerular filtration rate (eGFR), an index of kidney function, 74 of the 211 proteins that were significantly associated with general cognitive function remained significantly associated with the outcome, and 72 of the 188 proteins remained significantly associated with DSST (Supplementary Data 19 and 20). Some biomarkers of kidney function that were associated with general cognitive function or DSST, such as beta-2-microglobulin (B2M) and cystatin C[16], were no longer significant after controlling for eGFR ($p > 0.05$). However, a Mendelian randomization analysis of eGFR on general cognitive function did not support a potential causal effect of kidney function on general cognitive function (inverse variance weighted multiplicative random effect [IVW MRE] beta = −0.01, $p = 9.37$E-1, Supplementary Note 1, Supplementary Table 1).

**Gene sets enriched for proteins associated with cognitive function**. Over-representation analysis identified 13 enriched Gene Ontology (GO) gene sets associated with general cognitive function using the discovery meta-analysis results among participants aged ≥25 (Fig. 3, Supplementary Data 21). Of these, 2 were also enriched using the discovery results of general cognitive function among participants aged ≥65 (Supplementary Fig. 9, Supplementary Data 22). Among the 13 enriched gene sets, 4 were involved in cytokine and chemokine binding and activity.

Gene Set Enrichment Analysis identified 8 enriched GO gene sets using the discovery meta-analysis results of general cognitive function among participants aged ≥65 (Supplementary Fig. 10, Supplementary Data 23). Four of these were involved in synaptic or postsynaptic organization, and two were involved in immunity. Four enriched GO gene sets were identified using the discovery meta-analysis results of DSST among participants aged ≥65 (Supplementary Fig. 11, Supplementary Data 24), including a gene set involved in immune response that was also identified for general cognitive function among participants aged ≥65. We did not identify any enriched pathways from the Kyoto Encyclopedia

of Genes and Genomes (KEGG) library using any of the three meta-analysis results.

**Proteins significantly affecting or affected by general cognitive function based on MR analysis**. In the MR analysis of protein effect on general cognitive function, 202 of the 246 proteins that were significant in the discovery meta-analysis had one or more *cis* proxy single nucleotide polymorphisms (SNP) available. MR analysis implicated two proteins, inactive tyrosine-protein kinase 7 (PTK7) and DnaJ homolog subfamily B member 12 (DNAJB12), had effects on general cognitive function (inverse variance weighted fixed effect [IVW FE] $p < 3.96$E-5, Supplementary Data 25). Each had only one genetic proxy (proxy SNP I2 in protein quantitative trait locus in *cis* [*cis*-pQTL] meta-analysis, DNAJB12: 8.2, PTK7: 85.7, Supplementary Data 26). In the results of the Genotype-Tissue Expression (GTEx) project[17], these proxy SNPs were significantly associated with the expression levels of multiple cis-genes (Supplementary Data 27). In a database of expression quantitative trait locus (eQTL) of the brain cortex[18], these proxy SNPs did not associated with the expression levels of the protein encoding gene at FDR < 0.05.

Colocalization analyses for these two proteins provided support for a single shared causal variant for DNAJB12 (posterior probability [PP] for H4 = 0.86, Supplementary Data 28, Supplementary Fig. 12) and weak support for a shared causal variant for PTK7 (PP H4 = 0.68, Supplementary Fig. 13). The sum of single effect (SuSiE) method, which supports multiple signals in colocalization analysis, did not identify any credible set in the region of the encoding gene of these two proteins for the genetic association of general cognitive function.

In the MR analysis of general cognitive function on protein levels, genome-wide proxy SNPs of general cognitive function were analyzed against 243 proteins as outcome after excluding 3 protein complexes (Methods). Of these, MR analysis indicated that general cognitive function had effects on one protein (SLIT and NTRK-like protein 3 [SLITRK3], IVW multiplicative random effect [MRE] $p = 2.45$E-6), with support from one or more MR methods robust to pleiotropy (Supplementary Data 29, Supplementary Fig. 14). The proxy SNPs of general cognitive function had modest heterogeneity in the pQTL meta-analysis of SLITRK3 (I2, median (1st, 3rd quartile): 0 (0, 35.4), Supplementary Data 30).

**Proteins significantly affecting or affected by AD susceptibility from MR analysis**. In the MR analysis of protein effect on AD susceptibility, 202 of the 246 proteins that were significant in the discovery meta-analysis had one or more cis proxy SNPs available. There was evidence that one protein (nectin cell adhesion

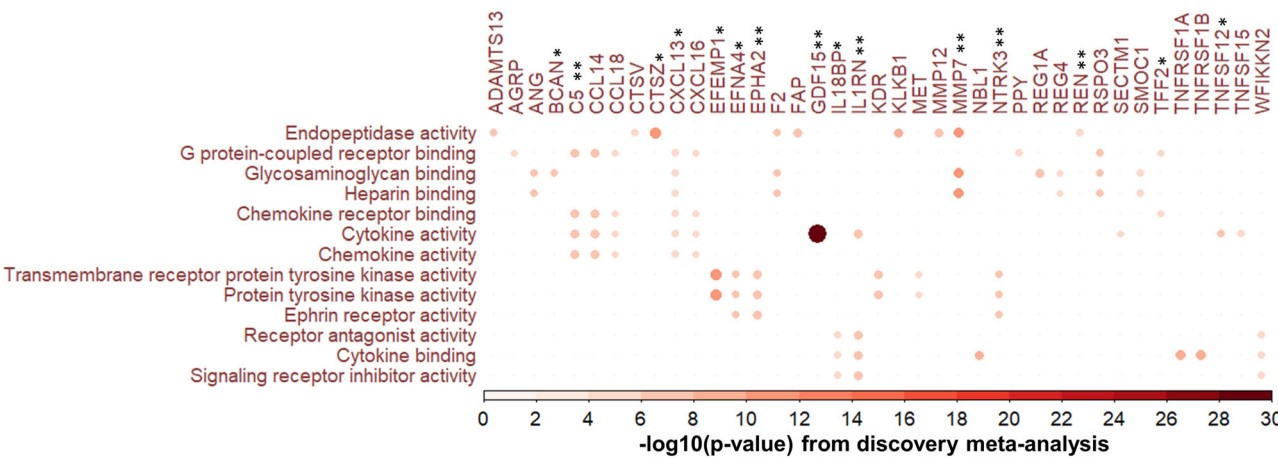

**Fig. 3 Gene Ontology (GO) terms that were enriched in the association between circulating proteins and general cognitive function in the discovery meta-analysis among those aged ≥ 25 based on overrepresentation analysis.** The proteins on the horizontal axis ($n = 39$) were those significant in the discovery meta-analysis and linked to the significant GO terms (Supplementary Data 21). Of these 15 were replicated: ** indicates those replicated at Bonferroni-corrected significance level ($n = 7$, Supplementary Data 15), and * indicates those replicated only at FDR < 0.05 ($n = 8$). The size and color of the circles correspond to –log10($p$-value) from the discovery meta-analysis (Supplementary Data 9).

**Table 2 Results of Mendelian randomization analysis relating cognition-associated proteins with Alzheimer's disease.**

| SomaScan ID | Entrez gene symbol / protein symbol | Uniprot ID | Uniprot name | # of proxy SNPs | Odds ratio | Beta[a] | SE | P-value** |
|---|---|---|---|---|---|---|---|---|
| Protein on AD | | | | | | | | |
| 6245_4 | NECTIN2 | Q92692 | Nectin-2 | 1 | 18.02 | 2.89 | 0.27 | 2.54E-26 |
| AD on protein | | | | | | | | |
| 4337_49 | CRP | P02741 | C-reactive protein | 17 | -- | −0.149 | 0.021 | 3.33E-12 |
| 4971_1 | CTSZ | Q9UBR2 | Cathepsin Z | 17 | -- | −0.051 | 0.009 | 2.11E-08 |
| 8039_41 | FAM177A1 | Q8N128 | Protein FAM177A1 | 17 | -- | 0.057 | 0.010 | 3.79E-08 |
| 10565_19 | SLITRK3 | O94933 | SLIT and NTRK-like protein 3 | 17 | -- | 0.033 | 0.006 | 1.53E-07 |
| 9348_1 | C1RL | Q9NZP8 | Complement C1r subcomponent-like protein | 17 | -- | −0.027 | 0.006 | 1.80E-05 |
| 9199_6 | UBE2G2 | P60604 | Ubiquitin-conjugating enzyme E2 G2 | 16 | -- | −0.038 | 0.009 | 6.74E-05 |
| 13950_9 | CERT1 | Q9Y5P4 | Collagen type IV alpha-3-binding protein | 17 | -- | −0.022 | 0.006 | 1.01E-04 |
| 9296_15 | PTPRD | P23468 | Receptor-type tyrosine-protein phosphatase delta | 17 | -- | 0.024 | 0.006 | 1.99E-04 |

*SE* standard error, *AD* Alzheimer's disease.
**$P$-value of NECTIN2 on AD susceptibility was obtained using Wald ratio since NECTIN2 only had one genetic proxy. *P*-value of AD susceptibility on protein was obtained using the inverse variance weighted multiplicative random effect method.
[a]The beta of protein on AD susceptibility was in the unit of log (odds ratio for AD) per SD of inverse normal transformed protein levels. The beta of AD susceptibility on protein was in the unit of SD of inverse normal transformed protein levels per log(odds ratio for AD).

molecule 2, NECTIN2) significantly increased the risk of AD (OR 18.0 per SD of protein levels, Wald ratio $p = 2.54E-26$, Table 2). This significant effect of NECTIN2 was also observed using two other summary statistics datasets with AD or AD-by-proxy as the outcome (Supplementary Data 31). NECTIN2 had one proxy SNP (rs440277, I2 in cis-pQTL meta-analysis: 39.1, Supplementary Data 26). This variant is in intron 1 of *NECTIN2*. Its G allele was significantly associated with higher circulating protein levels and higher gene expression levels of NECTIN2 in the liver, pancreas, and whole blood in the results from GTEx ($p < 1e-30$, Supplementary Data 32). This variant (rs440277) was also associated with the expression of AC084219.4, a long non-coding RNA, in the pancreas with much weaker effect ($p = 2.08E-05$). These lookup results support rs440277 as a specific proxy for circulating NECTIN2. In an eQTL dataset of the brain cortex, rs440277 was not associated with the expression levels of *NECTIN2* at FDR < 0.05[18]. Although the encoding genes of *NECTIN2* and *APOE* are in close proximity (26 kb), the correlations

between rs440277 and the two *APOE* ε4 variants were low (1000 G EUR: r2 = 0.018 with rs429358 and 0.001 with rs7412) with moderate D', a measure of linkage disequilibrium (1000 G EUR: D' = 0.42 with rs429358 and 0.11 with rs7412). We also replicated a previous finding of higher levels of SVEP1 (SomaScan ID: 11109_56 and 11178_21) as potentially causal for AD susceptibility (IVW FE $p = 3.28E-3$ and 3.76E-3, respectively, Supplementary Data 33)[9].

Colocalization analyses within the 500 kb region on both sides of the *NECTIN2* promoter, which included the *APOE* gene, did not support a single shared causal variant underlying NECTIN2 protein levels and AD susceptibility (PP of $H_4$: 2.0E-11, Supplementary Data 28), but did support the $H_3$ hypothesis that both traits are associated, but with different causal variants ($H_3 > 0.9$) (Supplementary Fig. 15). An analysis using SuSiE also did not support shared causal variants between NECTIN2 and AD susceptibility (all pairwise $H_4$ PP < 3.5E-04, Supplementary Data 34).

For the analysis of AD susceptibility on protein levels, genome-wide proxy SNPs of AD susceptibility were analyzed against 243 proteins as outcome after excluding 3 protein complexes (Methods). Using clinically diagnosed AD as the exposure (Kunkle et al. stage 1 dataset)[19], we identified 8 proteins as potentially affected by AD susceptibility (IVW MRE $p < 1.99E-4$, Table 2, Supplementary Data 35, Supplementary Figs. 16–23). Of these, 7 were supported by one or more methods robust to pleiotropy at $p < 0.05$ (Supplementary Data 36). In individual-level data, these protein measures were modestly correlated (abs(Pearson correlation of log2 transformed values, median [1st and 3rd quartile]: 0.13 [0.08, 0.18], Supplementary Data 37). The strongest effect was on C-reactive protein (CRP) with higher AD susceptibility leading to lower CRP levels (beta: −0.15 SD, IVW MRE $p = 3.33E-12$). In the analysis using 2 proxy SNPs inside the APOE region, 4 of the 8 proteins remained significant (CRP, CTSZ, FAM177A1, and UBE2G2), and none were significant using 15 proxy SNPs outside of the APOE region (IVW MRE $p > 1.6E-3$, Supplementary Data 35).

Using both clinically diagnosed AD and AD-by-proxy as the exposure (Jansen et al. dataset)[20], we identified 8 proteins as potentially affected by AD susceptibility (Supplementary Data 38, Supplementary Figs. 24–31). All these results were also supported by one or more methods robust to pleiotropy at $p < 0.05$ (Supplementary Data 39). In individual-level data, these protein measures were modestly correlated (abs(Pearson correlation of log2 transformed values, median [1st and 3rd quartile]: 0.17 [0.11, 0.23], Supplementary Data 37). Of these 8 significant proteins, two (CTSZ and FAM177A1) were the same as those identified using clinically diagnosed AD as exposure (Kunkle et al. stage 1 dataset) with the same effect direction[19]. In the analysis using proxy SNPs inside of the APOE region, 4 of the 8 proteins remained significant, and none were significant using proxy SNPs outside of the APOE region (IVW MRE $p > 8.69E-2$, Supplementary Data 38). The proxy SNPs of AD had modest heterogeneity in the pQTL meta-analysis (I2, median (1st, 3rd quartile): 0 (0, 16.4), Supplementary Data 30).

Interestingly, among the 14 unique protein measures that were significantly affected by AD susceptibility using the Kunkle et al. or Jansen et al. datasets, the effects from the MR analysis were in an unexpected direction from their associations with cognitive function in the discovery analysis (Supplementary Data 40). Specifically, given that higher AD susceptibility is associated with lower general cognitive function (genetic correlation: −0.37)[21], when higher AD susceptibility was associated with higher protein levels, it would be expected that higher levels of these proteins would be associated with lower cognitive function. However, in our discovery analysis, higher levels of these proteins were associated with higher cognitive function (Supplementary Data 40). This unexpected direction was also observed when higher AD susceptibility was associated with lower protein levels. For example, MR analysis suggested higher AD susceptibility led to lowered CRP levels as reported above (beta: −0.15 SD, IVW MRE $p = 3.33E-12$). Based on this MR result, it would be expected that lower CRP levels would be associated with lower cognitive function. However, in the discovery meta-analysis, higher CRP levels were associated with lower cognitive function (beta = −0.047 SD, $p = 2.2E-6$, Supplementary Data 40). To further investigate this unexpected direction of effect, we evaluated the association between APOE ε4 carrier status and the levels of the 14 proteins implicated to be affected by AD susceptibility given that APOE ε4 has been the genetic locus with the largest effect size on AD, and the genetic variants in APOE region were used in the MR analysis[19]. Based on data from ARIC and CHS, among 13 of the 14 proteins, the association between the APOE ε4 carrier status and these proteins support the effect direction from the MR analysis (binomial $p = 4.0E-3$, Supplementary Data 40). For example, APOE ε4 carriers had lower CRP levels (effect −0.33 SD, $p = 2.54E-22$). These results suggest that the potential effects of the APOE region on these proteins and cognitive function might act through different pathways.

**Results of additional analysis on the potential effect of AD susceptibility on CRP levels.** A lookup of the Catalog of human genome-wide association studies (GWAS) on the association between the APOE ε4 variants and CRP in large-scale biobank studies confirm that the risk alleles of APOE ε4 variants were associated with lower CRP levels, consistent with results from MR analysis (Supplementary Data 41)[22]. To gain additional insight into the effect of AD susceptibility on CRP levels, we used MR analysis to evaluate the effect of AD susceptibility on IL6, a regulator of CRP expression[23]. AD susceptibility overall or from the APOE region did not have significant effects on IL6 levels as measured by SomaScan based on Bonferroni-corrected threshold (IVW MRE genome-wide beta = −0.019, $p = 5.7E-2$; APOE region beta = −0.024, $p = 3.9E-1$). Regarding CRP, using data from participants in the ARIC study in the discovery analysis, we confirmed that the correlation between measures of CRP using SomaScan and a high-sensitivity assay was high (Pearson $\rho = 0.94$ after natural log transformation). In addition, using data from the participants from ARIC and CHS in the discovery analysis, we observed that higher CRP levels were associated with lower general cognitive function among both APOE ε4 carriers and non-carriers (ε4 carriers: beta = −0.071, $p = 1.3E-4$, $n = 1730$; non-carriers, beta = −0.037, $p = 2.0E-3$, $n = 4841$, $p$ for interaction 0.06). These results suggest the association of CRP as an inflammation marker with cognitive function is independent of the effect of APOE ε4 on CRP levels.

## Discussion

We identified 246 circulating proteins associated with general cognitive function or performance in DSST using data from 3 population-based cohorts. Of these, 45 were externally replicated using measures from the same assay platform as the discovery analysis (SomaScan), and 3 using measures from a complementary assay platform (Olink) at Bonferroni-corrected significance. The cognitive function-associated proteins were enriched in gene sets involved in chemokine and cytokine signaling, immune response, and synapse architecture. Two-sample Mendelian randomization analysis identified that higher levels of DNAJB12 and PTK7 could potentially lead to higher cognitive function, and higher levels of NECTIN2 could potentially increase AD susceptibility.

The results that the cognition-associated proteins were enriched in gene sets involved in chemokine and cytokine signaling, immune response, and synapse architecture fit with the findings that proteins involved in chemokines and cytokine signaling were associated with risk of dementia[24]. The signaling gene sets included members of the tumor necrosis factor and interleukin superfamilies. These results are consistent with the neuroinflammation and immune system pathways being implicated in AD risk[25,26]. Furthermore, inhibition of tumor necrosis factors has been associated with reduced risk for dementia[27]. The enriched gene sets involved in synapse architecture included postsynaptic density assembly, postsynaptic specialization assembly, excitatory synapse assembly, and postsynaptic density organization. These results support previous findings of risk factors for dementia, particularly AD, where dysregulation of postsynaptic junctions is a marker of loss of neuronal plasticity[28].

Two proteins (DNAJB12 and PTK7) were implicated as affecting general cognitive function based on MR analysis. The

effect of DNAJB12 was supported by colocalization analysis. This protein belongs to the heat shock protein family and is involved in protein degradation[29]. The cis-pQTLs of these proteins were not significant eQTLs of these proteins in brain tissues suggesting that the protein risk factors for cognitive function may differ between tissues. The cis-pQTLs of these two proteins were not significant eQTL of these proteins in brain tissues suggest that the protein risk factors for cognitive function may differ between tissues. However, given that the genetic proxies of DNAJB12 and PTK7 were associated with expression of other genes in the promoter regions of the protein encoding genes, the effect of the genetic proxies used in the MR analysis might reflect the effect of other proteins that are encoded in the same region and were not measured in our study.

Our MR analysis of AD susceptibility identified higher levels of NECTIN2 as potentially causal for AD susceptibility. NECTIN2 mediates viral entry into cells and modulates T-cell signaling[30]. NECTIN2 is found at the adhesion sites between endothelial cells in blood vessels and astrocytes, and has been implicated in age-related loss of neurons[30]. The potential causal effect of NECTIN2 on AD susceptibility is consistent with the hypothesis that infectious agents may trigger higher production of amyloid beta and thus aggravate AD pathology[31]. Infections, including influenza and pneumonia were found to be associated with incident AD or dementia in large-scale studies from multiple countries[32–35]. Variants and haplotypes at *NECTIN2* have been associated with AD independent of the *APOE* ε4 allele[36,37]. *NECTIN2* knockout mice were reported to show degeneration of astrocytic perivascular end foot processes and neurons in the cerebral cortex[38]. However, the MR effect of NEC-TIN2 on AD susceptibility might partly reflect the linkage disequilibrium between the genetic proxy of NECTIN2 and an *APOE* ε4 variant. Colocalization analysis of the genetic associations of circulating NECTIN2 and AD at the *NECTIN2* promoter region did not support the hypothesis that NECTIN2 levels and AD risk share a single causal variant in that region. However, the 500 kb region around the *NECTIN2* gene also included *APOE*. The extremely strong association of the APOE ε4 variant (rs429358) with AD and the imprecision of imputation might have resulted in difficulties in identifying independent signals in this region[39]. Therefore, NEC-TIN2 remains of high interest for future studies related to cognitive function.

Regarding AD susceptibility on circulating protein levels, our strongest finding was that AD susceptibility from the *APOE* region resulted in lower levels of CRP, a biomarker of chronic inflammation[40]. Multiple observational studies from different ancestries have reported that the *APOE* ε4 allele was associated with lower CRP levels[41–45]. These results appear to be counterintuitive given that both *APOE* ε4 allele and higher CRP levels have been associated with lower cognitive function, one would expect that *APOE* ε4 allele would be associated with higher CRP levels. The results of the MR analysis suggests that the effect of *APOE* ε4 allele on lower CRP levels is likely causal. An effect estimate from an MR analysis represents a lifetime effect, which could be different from the effect of the APOE protein during the preclinical or clinical stages of AD. Given that CRP is only one of many markers of inflammation, if the APOE ε4 allele indeed leads to lower CRP levels, this does not necessitate that other inflammatory markers or inflammation in general are lower among APOE ε4 carriers. Observational studies have also shown that higher CRP levels were associated with lower cognitive function[46]. This is consistent with the results from our discovery and replication analysis and with the analysis stratified by APOE ε4 carrier status. The association of higher CRP levels with lower cognitive function may represent the effect of inflammation, of which CRP is a biomarker, on cognitive function among both ε4 carriers and non-carriers regardless of their difference in lifelong CRP levels. Among the proteins that were

implicated as affected by AD susceptibility were two lysosomal cysteine proteinase (CTSA and CTSZ) and an ubiquitin conjugating enzyme (UBE2G2), proteins that are involved in ubiquitin signaling and lysosomal function. These pathways are known to be dysregulated in AD[47,48].

The proteins that were replicated using a complementary assay platform (Olink) were known to be associated with neural development, immune response, and cell signaling. For example, EFNA4 is thought to mediate nervous system development and hippocampal potentiation[49] and belongs to the protein tyrosine kinase activity gene set, which is significant in the enrichment analysis. The inflammatory marker CDCP1 has been associated with AD risk previously[50], and macrophage scavenger receptor types I and II (MSR1) is thought to bind β-amyloid and participate in its clearance[51].

Our sensitivity analysis adjusting for kidney function showed that the association of some protein measures were attenuated. However, two large-scale MR studies reported no evidence supporting the causal effect of kidney function on dementia or AD[52,53]. Our MR analysis also did not support kidney function as a potential causal factor of general cognitive function. A large-scale proteomic study using the SomaScan platform only implicated one protein out of almost 5000 as a potential causal factor of kidney function and suggested that most proteins are likely markers of kidney function[54]. Therefore the attenuation of the association between protein measures and cognitive function after adjusting for eGFR was likely due to statistical correlation between eGFR and protein measures rather than causal relationship between kidney function and cognitive function.

The strengths of this study include the use of multiple population-based studies in both discovery and replication analyses. Given the heterogeneity of tests used for cognitive assessment among cohort studies, we used a principal component-based approach to combine multiple cognitive tests into a measure of general cognitive function as done previously in GWAS of general cognitive function[3]. Some limitations warrant mentioning. Two of the discovery cohorts (CHS and FHS Gen 3) had a time gap between blood drawn for protein assay and cognitive assessment. However, there was a high level of consistency in the association results between CHS and ARIC, which did not have a time gap between blood drawn for protein assay and cognitive assessment. Among the protein measures that were significant in the discovery analysis, some had considerable heterogeneity, which may partly reflect different protein levels between middle age and older age[55]. We also had limited sample size for replication. These limitations represent the challenges in the current state of proteomics studies, including heterogeneity between assay platforms and protein levels in subpopulations[8,55]. In two-sample Mendelian randomization analysis, the results might be biased toward false negative when the datasets for the exposure and the outcome did not overlap, such as in our case[56]. However, given that we used strong genetic proxies (SNPs with genome-wide significance), the bias is likely small[56]. For the MR analysis using <3 proxy SNPs, there is a lack of methods for evaluating potential pleiotropy. Finally, the MR estimates based on the datasets that included AD by proxy as cases may be biased and did not adequately represent AD risk.

In conclusion, we identified circulating proteins associated with cognitive function across several independent cohorts, implicated higher NECTIN2 levels as increasing AD risk, and the potential role of AD susceptibility from the *APOE* region in regulating some circulating protein levels.

## Methods

**Study population in discovery and replication analyses.** The participant inclusion criteria were aged ≥25, without prevalent

dementia and stroke, and having data for proteomics, cognitive function, and covariates. The discovery analysis included three population-based cohorts in the Cohorts for Heart and Aging Research in Genomic Epidemiology (CHARGE) consortium: the Atherosclerosis Risk in Communities (ARIC) study, Cardiovascular Health Study (CHS), and Framingham Heart Study (FHS) Gen 3[57,58]. These discovery cohorts contributed protein measures quantified using the SomaScan assays from SomaLogic (Boulder, CO) (Supplementary Note 2). The primary replication analysis included four cohorts using protein measures quantified using the SomaScan assay (the Age, Gene/Environment Susceptibility – Reykjavik Study [AGES], the Baltimore Longitudinal Study for Aging [BLSA], the Multi-Ethnic Study of Atherosclerosis [MESA], and the Whitehall II study). The exploratory replication analysis included five cohorts using the immunoassay platform from Olink (Uppsala, Sweden) (the Coronary Artery Risk Development in Young Adults [CARDIA] study, the Lothian Birth Cohort [LBC] 1921, and LBC1936, the Rhineland Study, and the Three-City Study [3 C]) (Supplementary Note 2). This study was conducted in accordance with the Declaration of Helsinki. All participants gave their informed consent for inclusion before they participated in the studies. The protocol of the studies have been approved by the respective institutional review board.

**Quantification of circulating proteins.** Technical variations of the protein assays were normalized based on manufacturer protocols (Supplementary Data 2). For protein measures quantified using the SomaScan platform, non-human proteins and proteins that were known to bind to contaminates or related proteins with similar affinity were excluded from the discovery meta-analysis (Supplementary Data 2).

**Outcome definitions.** The primary outcome was general cognitive function, represented by the first unrotated principal component (PC1) of cognitive scores from at least three different domains (Supplementary Data 6). The principal component scores were oriented in the direction of higher cognitive function. This approach has been used by previous GWAS for discovering associations between genetic variants and general cognitive function[3]. The variance of the cognitive test scores explained by PC1 ranged from 0.47 to 0.67 among discovery and replication cohorts (Supplementary Data 7). The contribution of each cognitive score to PC1 (loading) and the correlation between each cognitive score and PC1 are reported in Supplementary Data 5 and 6. We also used the score from the DSST, a test assessing processing speed, as a separate outcome. The DSST was used by two discovery cohorts and was included in the generation of general cognitive function in these cohorts (Supplementary Data 5).

**Discovery and replication analyses of the association between proteins and cognitive function.** A flow chart of the main analyses is presented in Fig. 1. Given that some proteins were measured by multiple aptamers on the SomaScan platform, the values ascertained by each aptamer are referred as protein measures and analyzed separately. In the discovery analysis within each cohort, the protein measures were transformed on the log2 scale to reduce skewness. Both the cognitive outcomes and protein measures were standardized to a mean of 0 and a standard deviation (SD) of 1. Each cohort conducted linear regression to evaluate the association between each protein as the independent variable and each cognitive outcome as the dependent variable controlling for age, sex, APOE ε4 carrier status, education attainment, and the number of days between blood drawn for protein assay and cognitive assessment if the blood draw and

assessment did not occur on, or very near, the same day. Multi-center and multi-ethnic cohorts also included center and self-reported race as covariates. The overall analysis included all participants aged ≥25 in the 3 discovery cohorts ($n = 7289$). An association analysis only including participants aged ≥65 ($n = 6583$) was also conducted given that the levels of some circulating proteins have large variations from middle age to older age and the association between circulating proteins and cognitive function may differ by age group[8,55]. This analysis only included data from the two discovery cohorts (ARIC and CHS) with participants aged ≥65.

In the discovery meta-analysis, we used the inverse variance weighted fixed effects method to combine the association statistics. Given that the meta-analysis of general cognitive function among those aged ≥25 only had one cohort with participants aged <65 and the meta-analysis among those age ≥65 only had two cohorts, we required that each meta-analysis included only protein measures that were present in all cohorts. The meta-analysis of general cognitive function among participants aged ≥25 included 1049 protein measures given that one discovery cohort (FHS) used the SomaScan 1 K platform, which only included 1305 aptamers. The meta-analysis of general cognitive function and DSST among participants aged ≥65 included 4709 protein measures from the two cohorts (CHS and ARIC) contributing results among participants aged ≥65 used SomaScan version 4, which included 5284 aptamers. The 1049 protein measures in the analysis among participants aged ≥25 were a subset of the proteins analyzed among participants aged ≥65 analysis. The statistical significance thresholds for discovery were Bonferroni corrected based on the number of protein measures in each analysis (aged ≥25: 4.9E-5 = 0.05/1049; aged ≥65: 1.1E-05 = 0.05/4709).

Protein measures that met the Bonferroni-corrected significance threshold and had a consistent direction of effect across all discovery cohorts included in each meta-analysis were selected for replication. The analysis methods within each replication cohort were the same as those in the discovery analysis, except that some cohorts employed a different transformation of protein values as noted in the supplemental materials. To combine the association statistics of the replication cohorts, we conducted an inverse variance weighted fixed effects meta-analysis separately for results using the SomaScan and Olink proteomic platforms for each outcome (general cognitive function vs DSST) and age group. We considered the replication analysis using SomaScan data as primary and the replication analysis using Olink data as exploratory given that the two platforms may not measure the same characteristics of a protein[14,59]. Our primary replication significance threshold is Bonferroni-corrected based on the number of proteins selected for replication and available among the replication cohorts in each analysis. We also report proteins with Benjamini-Hochberg false discovery rate (FDR) < 0.05. We performed *post-hoc* power calculation to estimate the power for replicating a protein measure based on its effect size from the discovery analysis, the replication sample size, and the Bonferroni-correct significance level given the number of protein measures selected and available for replication using the pwr.f2.test function in R. These calculations were conducted for protein measures available for replication from the SomaScan and Olink platforms separately. The correspondence between proteins available in the SomaScan and the Olink platforms was identified using the UniProt ID in the protein annotation provided by the respective manufacturers[60]. Meta-analyses were conducted using the metafor package in R[61].

**Exploration of relationships between proteins replicated on the SomaScan platform.** To explore potential relationships among protein measures replicated at Bonferroni-corrected significance

using the SomaScan platform, we plotted the pairwise Pearson correlations between these measures in each analysis after applying hierarchical clustering based on Euclidean distance of the correlations using the R package heatmaply version 0.14.1[62]. We also plotted the potential interactions between the replicated proteins in each analysis using data from the STRING database (version 12)[15].

**Sensitivity and other analyses of cognition-associated proteins**. For the protein measures that were significant in the discovery study, we interrogated a large-scale GWAS of general cognitive function for genetic variants that were associated at $p < 5E-8$ and located at 500 kb on both sides of the promoter of the encoding gene[3]. We also queried the results of the associations between proteins and incident dementia conducted in the ARIC study, one of discovery cohorts, and the results of a proteomic study on cognitive decline and general cognitive ability[7,9,13]. To investigate the agreement in quantification between assay platforms, we queried the correlation between protein levels from the SomaScan and the Olink platforms in publicly available results[14].

Given that kidney function may explain a substantial proportion of the variance of circulating protein levels among older adults, who tend to have lower kidney function[63], we conducted sensitivity analysis of the association between the protein measures and cognitive function in the two discovery cohorts (ARIC and CHS) with participants aged ≥65 additionally controlling for kidney function, represented by eGFR based on a recent equation using serum creatinine and cystatin C[64]. Kidney function was not included in the primary discovery model because some circulating proteins widely considered as biomarkers of kidney function, such as beta 2 macroglobulin (B2M) and cystatin C, have been reported to have biological function in the brain[65,66].

**Enrichment analysis**. Enrichment analysis identifies whether any gene sets or pathways were enriched with cognitive function-associated proteins. For the over-representation analysis, we used proteins that were significant at a Bonferroni-corrected $p$-value of 0.05 in the discovery meta-analysis and all tested proteins as the background. We also employed Gene Set Enrichment Analysis (GSEA), which determines whether members of a gene set were enriched toward the top or bottom of the effect estimate distribution[67]. The candidate gene sets or pathways were from the GO and KEGG libraries[68,69]. We tested for the gene sets or pathways with 10 to 200 genes. The enrichment significance threshold was FDR < 0.05. Analysis was performed using ClusterProfiler package (version 4.4.4) in R version 4.2.1[70].

**Mendelian randomization (MR) analysis of significant proteins from the discovery meta-analysis**. For the protein measures that had Bonferroni-corrected significant associations with either general cognitive function or DSST in our discovery meta-analysis, we conducted two-sample bi-directional MR analysis to evaluate whether these proteins may affect cognition-related outcomes or vice versa. The summary statistics of GWAS of proteins were from a fixed-effect meta-analysis of three studies of pQTL among individuals of European ancestry with the protein measures quantified using the SomaScan platform (total sample size up to 49,376)[71–73]. The meta-analysis was performed using metal with the rs number as the SNP identifier. GWAS for cognition-related outcomes were Alzheimer's disease (AD) and general cognitive function among European populations excluding cohorts from the CHARGE consortium (Supplementary Note 2)[3,19,20,74]. The populations included in the pQTL summary

statistics and those of the cognition-related summary statistics did not overlap.

An MR analysis uses genetic proxies of an exposure to evaluate the association between the genetically predicted levels of the exposure and an outcome. If the association is significant and the genetic proxies of the exposure satisfy three MR assumptions of being valid genetic proxies, then one could infer that the MR effect estimate represents the lifetime causal effect of the exposure on the outcome. The three MR assumption are: (1) the genetic proxy is associated with the exposure, (2) the genetic proxy is independent of measured or unmeasured confounders, and (3) the genetic proxy can only influence the outcome through the exposure, that is, it cannot have a pleiotropic effect on the outcome independent of the exposure[75].

We considered the above three MR assumptions in deciding our selection criteria for genetic proxies. To address the assumption of the genetic proxy being associated with the exposure, we selected genetic variants that were associated with an exposure at genome-wide significance ($p$-value < 5E-8). Genetic proxies are limited to those with minor allele frequencies (MAF) > 1%, non-palindromic, and independent (r2 < 0.001 based on 1000 Genomes EUR reference panel). To reduce potential reverse causation, we applied Steiger filtering to exclude genetic proxies that explained a larger proportion of the variance of the outcome than that of the exposure at $p < 0.05$[76].

When a protein measure was the exposure, to reduce potential confounding and pleiotropy, the genetic proxies for protein measures were limited to those located within 500 kb of the promoter of the encoding gene of the protein[77]. We excluded measures annotated to a pseudogene (BAGE2), which did not have promoter information, and protein complexes (C5/C6, CGA/FSHB, and CGA/LHB) because the promoter regions of these encoding genes were hard to define. In addition, genetic proxies for protein measures were required to exist in at least two of the three pQTL datasets to ensure that the association between the genetic proxy and the protein measure was not driven by only one pQTL dataset.

For the primary analysis methods, when a protein measure had only one genetic proxy, we used Wald ratio. When a protein measure had two genetic proxies, we used Wald ratio followed by fixed effect meta-analysis to combine the estimates from the two genetic proxies. When a protein measure was the exposure and had three or more genetic proxies, we used the fixed effects inverse variance weighted (IVW FE) method because the genetic proxies came from one genomic region[78]. When AD susceptibility or general cognitive function was the exposure, the primary method was the inverse variance weighted multiplicative random effect (IVW MRE) method because the genetic proxy came from multiple regions of the genome[78]. To assess potential heterogeneity, we used the Cochran's Q statistic[78]. To evaluate the effect of an exposure in the presence of potential pleiotropic effect of the genetic proxies, we used methods robust to pleiotropy (Egger regression, weighted median, weighted mode) as secondary methods[79–82]. These methods can provide valid causal estimates even when some proxy SNPs of the exposure may influence the outcome through pathways outside of the exposure and thus are not valid proxies of the exposure due to their pleiotropic effects.

For the analyses of general cognitive function or AD susceptibility on protein levels, we excluded the 3 protein complexes listed above. For protein measures whose levels were implicated as influenced by AD susceptibility, we further performed separate analyses using genetic proxies in the APOE region (defined as 250 Kb on both sides of the APOE gene) and those outside of the APOE region to evaluate whether the effects of AD susceptibility on these proteins largely originated from the APOE region. The significance level for the MR results was

Bonferroni-corrected $p$-value < 0.05 from the primary method. For proteins that were previously reported as having causal effects on dementia and were also significant in our discovery study, the significance threshold was $p$-value < 0.05 from the primary method. The MR analysis was conducted using the TwoSampleMR package[83].

Given that our sensitivity analysis of the association between protein measures and general cognitive function adjusting for kidney function showed that the association of some proteins attenuated, we considered whether kidney function might be a causal factor for general cognitive function and consequently a confounder or mediator in the relationship between protein and cognitive function. We conducted an MR analysis of eGFR estimated using serum creatinine on general cognitive function. The MR methods followed those described above. In addition, given that some SNPs that are strongly associated with eGFR may be proxy of serum creatinine instead of kidney function, we followed published methods to use the association of the SNP with blood urea nitrogen (BUN) to filter out SNPs that likely represented association with serum creatinine rather than kidney function[84–86]. We requires that a proxy SNP of eGFR was also associated with BUN in opposite effect direction (higher kidney function, lower BUN levels) at $p$-value < 0.05.

**GWAS datasets used in two-sample Mendelian randomization (MR) analyses**. The GWAS summary statistics of general cognitive function were from a meta-analysis of the summary statistics from the Cognitive Genomics Consortium (COGENT) and the UK Biobank ($n = 265,014$)[3] and did not include cohorts in the protein discovery analysis of cognitive function. The study populations of these cognition-related outcomes were also of European ancestry. The populations of the pQTL summary statistics and those of the cognition-related summary statistics did not overlap.

The primary GWAS summary statistics of AD were from the stage 1 meta-analysis of Kunkle et al. ($n = 21,982$ clinically diagnosed late-onset AD cases, 41,944 cognitively normal controls)[19]. To assess the generalizability of the primary MR analysis of protein on AD susceptibility, significant MR findings of proteins on AD using the Kunkle et al. dataset were further analyzed in two additional sets of summary statistics: the discovery summary statistics of Bellenguez et al. ($n = 20,464$ clinically diagnosed late-onset AD cases and 22,244 cognitively normal controls) which excluded the APOE region, and Phase 3 analysis of Jansen et al. ($n = 359,856$), whose cases overlapped with the Kunkle et al. dataset and included participants with clinically diagnosed late-onset AD and those having a parent with AD (AD-by-proxy cases)[20,74]. The Jansen et al. dataset was also used to identify proteins whose levels might be affected by AD susceptibility.

The eGFR GWAS summary statistics were the results from Stanzick et al. 2021 (European ancestry $n = 1,004,040$)[86]. The BUN summary statistics were from Wuttke et al. 2019 ($n = 243,031$)[85].

**Colocalization analysis of proteins with significant MR result on AD susceptibility**. Given that colocalization has been recognized as a complementary approach for inferring whether a trait may be causal to a disease[87], for protein measures with significant MR results on general cognitive function or AD susceptibility, we performed colocalization analysis to assess whether the genetic association of the exposure and outcome in the region of the protein encoding gene can be attributed to the same causal variant[88]. The region was defined as within 500 kb on both sides of the promoter. The SNP association statistics were the same as

those used in the MR primary analysis. The first method we used had the assumption that both traits only have one causal variant in the region[89]. Since there may be potentially multiple causal variants at a locus, we also used Sum of Single Effects (SuSiE) analysis[90,91], which applied variable selection method in regression to identify independent signals in the association of the SNPs with the exposure and outcome separately. Then for each pair of independent signals from the two traits, SuSiE applied colocalization analysis to estimate the posterior probability of having a shared causal variant (PP-H$_4$). The reference panel used for the SuSiE analysis was 1000 Genomes EUR. We performed colocalization analyses using the *coloc* R package and used a PP-H$_4$ > 80% to conclude that both traits shared the same causal variant.

**Query of the association of proxy SNPs of protein measures with gene expression**. For protein measures with significant effects on AD susceptibility or general cognitive function from MR analysis, we queried the associations of their proxy SNPs with gene expression in two datasets to determine whether these proxy SNPs might be associated with the expression of the protein-encoding gene. The first dataset was the tissue-specific expression of cis-genes in the GTEx project (version 8)[17]. The second dataset consisted of cortex-specific gene expression data[18]. For proxy SNPs of protein measures that were located near APOE, we looked up the linkage disequilibrium of the proxy SNP with the SNPs that form the APOE4 haplotype (rs429358 and rs7412).

**Additional analysis on the proteins potentially affected by AD susceptibility**. To explore the relationship between AD susceptibility from the *APOE* region and the protein measures that were significant in the MR analysis, we used linear regression to evaluate the association of *APOE* ε4 carrier status as the independent variable and each protein measure as the outcome controlling for age, sex, education attainment, race-center in ARIC and CHS followed by using fixed effect inverse variance weighted method to combine the results. The populations were the same as those used in the discovery analysis of cognitive function.

**Additional analysis on the relationship between AD susceptibility and CRP**. To evaluate whether the effect of AD susceptibility on CRP levels is consistent with the association between the APOE ε4 variants and CRP, we interrogated the GWAS catalog for these association in studies including large-scale biobanks[22]. Given that our MR analysis showed AD susceptibility from the APOE region led to lower CRP levels, as measured by the SomaScan platform and CRP is a biomarker of inflammation and has been associated with cognitive impairment[40,46,92], we conducted three additional analyses to further our understanding of this relationship between the APOE region and CRP. The first assessed the correlation between CRP levels as measured by the SomaScan assay and high-sensitivity immunoassay in the ARIC study. The second used MR analysis to investigate whether AD susceptibility from the APOE region affected the levels of interleukin 6 (IL6), a known regulator of CRP[93]. The MR methods were the same as those described above. The third evaluated the association between CRP, as measured by SomaScan, and general cognitive function in the ARIC study and CHS stratified by APOE ε4 carrier status using the same method as in the discovery analysis, except that the APOE ε4 carrier status became a stratifying variable instead of a covariate. The stratified results from ARIC and CHS were combined using inverse variance weighted fixed effect meta-analysis.

**Reporting summary**. Further information on research design is available in the Nature Portfolio Reporting Summary linked to this article.

## Data availability

The summary statistics of the discovery and replication meta-analysis of the association between circulating protein and cognitive function, including the sensitivity analysis controlling for kidney function, are available in figshare (https://doi.org/10.6084/m9.figshare.24069354). Source data underlying Fig. 3 are reported in Supplementary Data 9, 15, and 21. The datasets used in Mendelian randomization were download from the studies that published those results: deCODE protein quantitative trait loci (pQTL) dataset https://www.decode.com/summarydata/, Interval pQTL dataset: the European Genotype Archive (accession number EGAS00001002555), the Fenland pQTL dataset: https://gwas.mrcieu.ac.uk/ and https://www.ebi.ac.uk/gwas/, the GWAS of Alzheimer's disease by Kunkle et al.: The National Institute on Aging Genetics of Alzheimer's Disease Data Storage Site (NIAGADS) accession NG00075, the GWAS of Alzheimer's disease by Jansen et al.: https://ctg.cncr.nl/software/summary_statistics. The gene expression quantitative trait loci (eQTL) data were download from the Genotype-Tissue Expression (GTEx) portal https://gtexportal.org/home/datasets, the GWAS of general cognitive function: requests can be sent to the chairs of the CHARGE and COGENT consortia.

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

## Acknowledgements

The ARIC study has been funded in whole or in part with Federal funds from the National Heart, Lung, and Blood Institute, National Institutes of Health, Department of Health and Human Services (contract numbers HHSN268201700001I, HHSN268201700002I, HHSN268201700003I, HHSN268201700004I and HHSN268201700005I), R01HL087641, R01HL059367 and R01HL086694; National Human Genome Research Institute contract U01HG004402; and National Institutes of Health contract HHSN268200625226C. Funding was also supported by 5RC2HL102419, R01NS087541 and R01HL131136. Neurocognitive data were collected by U01 2U01HL096812, 2U01HL096814, 2U01HL096899, 2U01HL096902, 2U01HL096917 from the NIH (NHLBI, NINDS, NIA and NIDCD). Infrastructure was partly supported by Grant Number UL1RR025005, a component of the National Institutes of Health and NIH Roadmap for Medical Research. The authors thank the staff and participants of the ARIC study for their important contributions. Infrastructure was partly supported by Grant Number UL1RR025005, a component of the National Institutes of Health and NIH Roadmap for Medical Research. The Cardiovascular Heath Study (CHS) research was supported by NHLBI contracts HHSN268201200036C, HHSN268200800007C, HHSN268201800001C, N01HC55222, N01HC85079, N01HC85080, N01HC85081, N01HC85082, N01HC85083, N01HC85086, 75N92021D00006; and NHLBI grants U01HL080295, R01HL087652, R01HL105756, R01HL103612, R01HL120393, R01HL085251, R01HL144483, and U01HL130114 with additional contribution from the National Institute of Neurological Disorders and Stroke (NINDS). Additional support was provided through R01AG023629, R01AG15928, and R01AG20098 from the National Institute on Aging (NIA). AEF is supported by K01AG071689. The Framingham Heart Study is conducted and supported by the National Heart, Lung, and Blood Institute (NHLBI) in collaboration with Boston University (Contract No. N01-HC-25195, HHSN268201500001I and 75N92019D00031). This work was also supported by grant R01AG063507, R01AG054076, R01AG049607, R01AG059421, R01AG033040, R01AG066524, P30AG066546, U01 AG052409, U01 AG058589 from from the National Institute on Aging and R01 AG017950, UH2/3 NS100605, UF1 NS125513 from National Institute of Neurological Disorders and Stroke and R01HL132320. AGES has been funded by NIA contracts N01-AG012100 and HSSN271201200022C, NIH Grant No. 1R01AG065596-01A1, Hjartavernd (the Icelandic Heart Association), and the Althingi (the Icelandic Parliament). M. R. Duggan, T. Tanaka, J. Candia, K. A. Walker, L. Ferrucci, L.J. Launer, O. Meirelles are funded by the National Institute on Aging Intramural Research Program. This study was funded, in part, by the National Institute on Aging Intramural Research Program. We thank the BLSA participants and staff for their participation and continued dedication.

The Coronary Artery Risk Development in Young Adults Study (CARDIA) is supported by contracts HHSN268201800003I, HHSN268201800004I, HHSN268201800005I, HHSN268201800006I, and HHSN268201800007I from the National Heart, Lung, and Blood Institute (NHLBI). The LBC1921 was supported by the UK's Biotechnology and Biological Sciences Research Council (BBSRC), The Royal Society, and The Chief Scientist Office of the Scottish Government. Genotyping was funded by the BBSRC (BB/F019394/1). LBC1936 is supported by the Biotechnology and Biological Sciences Research Council, and the Economic and Social Research Council [BB/W008793/1], Age UK (Disconnected Mind project), and the University of Edinburgh. Genotyping was funded by the BBSRC (BB/F019394/1). The Olink® Neurology Proteomics assay was supported by a National Institutes of Health (NIH) research grant R01AG054628. Phenotype harmonization, data management, sample-identity QC, and general study coordination of the MESA were provided by the TOPMed Data Coordinating Center (3R01HL-120393-02S1), and TOPMed MESA Multi-Omics (HHSN2682015000031/HSN26800004). The MESA projects are conducted and supported by the National Heart, Lung, and Blood Institute (NHLBI) in collaboration with MESA investigators. Support for the Multi-Ethnic Study of Atherosclerosis (MESA) projects are conducted and supported by the National Heart, Lung, and Blood Institute (NHLBI) in collaboration with MESA investigators. Support for MESA is provided by contracts 75N92020D00001, HHSN268201500003I, N01-HC-95159, 75N92020D00005, N01-HC-95160, 75N92020D00002, N01-HC-95161, 75N92020D00003, N01-HC-95162, 75N92020D00006, N01-HC-95163, 75N92020D00004, N01-HC-95164, 75N92020D00007, N01-HC-95165, N01-HC-95166, N01-HC-95167, N01-HC-95168, N01-HC-95169, UL1-TR-000040, UL1-TR-001079, UL1-TR-001420, UL1TR001881, DK063491, and R01HL105756. The authors thank the other investigators, the staff, and the participants of the MESA study for their valuable contributions. The Rhineland Study is funded by the German Center for Neurodegenerative Disease (DZNE). This work was further partly supported by the German Research Foundation (DFG) under Germany's Excellence Strategy (EXC2151-390873048) and SFB1454 - project number 432325352; the Federal Ministry of Education and Research under the Diet-Body-Brain Competence Cluster in Nutrition Research (grant numbers 01EA1410C and 01EA1809C) and in the framework "PreBeDem - Mit Prävention und Behandlung gegen Demenz" (grant numbers 01KX2230); and the Helmholtz Association under the Initiative and Networking Fund (grant number RA-285/19) and the 2023 Innovation Pool. The 3 C Study is supported by a grant overseen by the French National Research Agency (ANR) as part of the "Investment for the Future Programme" ANR-18-RHUS-0002. It has received funding from the European Union's Horizon 2020 research and innovation programme under grant agreement No 667375. The project also received funding from the French National Research Agency (ANR) through the VASCOGENE and SHIVA projects and from the Leducq TNE 2012 on small vessel disease (PI A Joutel, M Nelson). Computations were performed on the Bordeaux Bioinformatics Center (CBiB) computer resources, University of Bordeaux. Funding support for additional computer resources has been provided to S.D. by the Fondation Claude Pompidou. The Three City Study: The Three City (3 C) Study is conducted under a partnership agreement among the Institut National de la Santé et de la Recherche Médicale (INSERM), the University of Bordeaux, and Sanofi-Aventis. The Fondation pour la Recherche Médicale funded the preparation and initiation of the study. The 3 C Study is also supported by the Caisse Nationale Maladie des Travailleurs Salariés, Direction Générale de la Santé, Mutuelle Générale de l'Education Nationale (MGEN), Institut de la Longévité, Conseils Régionaux of Aquitaine and Bourgogne, Fondation de France, and Ministry of Research–INSERM Programme "Cohortes et collections de données biologiques." Ilana Caro received a grant from the EUR digital public health. This PhD program is supported within the framework of the PIA3 (Investment for the future). Project reference 17-EURE-0019.

## Author contributions

Bioinformatics: A.T., I.C., J.V.L., JuC., M.P., O.M., V.E. Critical review of manuscript: A.H., A.M.T., A.T., A.S.B., A.S., A.E.F., A.C., A.L.F., B.M.P., C.L., C.L.S., C.M.S., C.T., D.L., E.M.T.D., G.D., J.B., J.I.R., J.Y., J.V.L., J.C., J.C.B., K.A.W., L.J.L., L.J., L.F., M.L., M.P., M.R.D., M.K., M.M.B.B., M.F., O.L.L., P.R., Q.Y., R.E.G., S.E.H., S.R.C., S.D., S.S.R., S.S., S.R.H., T.H.M., T.R.A., T.T., R.S.V., V.G.G., X.G. Drafting of manuscript: A.T., A.E.F. Interpretation of results: A.T., A.E.F., D.L., J.V.L., J.C., J.C.B., L.J.L., M.K., M.M.B.B., M.F., Q.Y., R.E.G., S.S., S.R.H., V.G.G. Management of an individual contributing study: A.M.T., A.E.F., A.L.F., B.M.P., J.C., J.C.B., J.I.R., K.A.W., M.K., M.M.B.B., P.R., S.R.C., S.D., S.S.R., S.S., S.R.H., T.H.M., R.S.V., V.G.G. Phenotyping: A.M.T., C.L., E.M.T.D., G.D., L.J., O.L.L. Statistical methods and analysis: A.T., A.S., A.E.F., A.C., B.M.P., C.M.S., D.L., E.W., G.D., I.C., J.Y., J.V.L., M.R.D., O.M., Q.Y., S.E.H., V.G., X.G., Y.Z. Study design of an individual study: C.L., C.L.S., J.V.L., J.C., K.A.W., L.J.L., L.J., L.F., M.K., M.M.B.B., R.E.G., S.E.H., S.R.C., S.D., S.S., T.M.H., R.S.V., V.G.G. Subject recruitment: A.M.T., A.S.B., J.C., L.F., M.M.B.B., O.L.L., S.R.C., S.D., T.H.M., R.S.V.

## Competing interests

E.W. is now an employee of AstraZeneca. M.K. is supported by the Wellcome Trust (221854/Z/20/Z), the UK Medical Research Council (MR/S011676/1), the US National Institute on Aging (R01AG056477), and the Academy of Finland (329202, 350426). B.P. serves on the Steering Committee of the Yale Open Data Access Project funded by Johnson & Johnson. J.C. serves as a Scientific Advisor to SomaLogic and receives grant support from NIH. Other coauthors have nothing to disclose. L.J. is an employee and stockholder of Novartis. All other authors declare no competing interests.

## Additional information

[1]Memory Impairment and Neurodegenerative Dementia (MIND) Center, University of Mississippi Medical Center, Jackson, MS, USA. [2]Department of Epidemiology, Johns Hopkins Bloomberg School of Public Health, Baltimore, MD, USA. [3]Department of Epidemiology, University of Washington, Seattle, WA, USA. [4]Institute for Public Health Genetics, University of Washington, Seattle, WA, USA. [5]Cardiovascular Health Research Unit, Department of Medicine, University of Washington, Seattle, WA, USA. [6]Department of Biostatistics, Boston University, Boston, MA, USA. [7]Lothian Birth Cohorts, Department of Psychology, University of Edinburgh, 7 George Square, Edinburgh EH8 9JZ, UK. [8]The Institute for Translational Genomics and Population Sciences, Department of Pediatrics, The Lundquist Institute for Biomedical Innovation at Harbor-UCLA Medical Center, Torrance, CA, USA. [9]Population Health Sciences, German Center for Neurodegenerative Diseases (DZNE), Bonn, Germany. [10]University of Bordeaux, Institut National de la Santé et de la Recherche Médicale (INSERM), Bordeaux Population Health Research Center, UMR 1219, CHU Bordeaux, Bordeaux, France. [11]Broad Institute of the Massachusetts Institute of Technology and Harvard University, The Klarman Cell Observatory, Cambridge, MA, USA. [12]Clinicum, Department of Public Health, University of Helsinki, Helsinki, Finland. [13]Department of Epidemiology and Public Health, University College London, London, UK. [14]Laboratory of Behavioral Neuroscience, National Institute on Aging, Baltimore, MD, USA. [15]National Institute on Aging, National Institutes of Health, Laboratory of Epidemiology and Population Science, Bethesda, MD, USA. [16]Faculty of Medicine, University of Iceland, Reykjavik, Iceland. [17]Icelandic Heart Association, Kopavogur, Iceland. [18]Institute of Cardiovascular Science, University of London, London, UK. [19]Framingham Heart Study, Framingham, MA, USA. [20]Department of Biostatistics,

University of Washington, Seattle, WA, USA. [21]Departments of Family Medicine, University of Washington, Seattle, WA, USA. [22]Precision Healthcare Institute, Queen Mary University of London, London, UK. [23]MRC Epidemiology Unit, University of Cambridge, Cambridge, UK. [24]Computational Medicine, Berlin Institute of Health at Charité – Universitätsmedizin Berlin, Berlin, Germany. [25]Department of Population Health Sciences and Glenn Biggs Institute for Alzheimer's & Neurodegenerative Diseases, UT Health San Antonio, San Antonio, TX, USA. [26]Department of Neurology, Boston University School of Medicine, Boston, MA, USA. [27]Department of Psychology, University of Texas, Austin, TX, USA. [28]Human Genetics Center, School of Public Health, University of Texas Health Science Center at Houston, Houston, TX, USA. [29]Johns Hopkins University, Baltimore, MD, USA. [30]Translational Gerontology Branch, National Institute on Aging, Baltimore, MD, USA. [31]Novartis Institutes for Biomedical Research, 22 Windsor Street, Cambridge, MA, USA. [32]McGill Genome Centre, Montreal, QC, Canada. [33]Departments of Neurology and Psychiatry, University of Pittsburgh, Pittsburgh, PA, USA. [34]Department of Medicine, Beth Israel Deaconess Medical Center, Boston, MA, USA. [35]Center for Public Health Genomics, Department of Public Health Sciences, University of Virginia, Charlottesville, VA, USA. [36]Department of Internal Medicine, Section of Gerontology and Geriatric Medicine, Wake Forest School of Medicine, Winston-Salem, NC, USA. [37]Department of Epidemiology and Prevention, Wake Forest University School of Medicine, Winston-Salem, NC, USA. [38]University of Texas School of Public Health in San Antonio, San Antonio, TX, USA. [39]University of Texas Health Sciences Center, San Antonio, TX, USA. [40]UCL Brain Sciences, University College London, London, UK. [41]Clinicum, Faculty of Medicine, University of Helsinki, Helsinki, Finland. [42]Institute for Medical Biometry, Informatics and Epidemiology (IMBIE), Faculty of Medicine, University of Bonn, Bonn, Germany. [43]Department of Neurology, Institute for Neurodegenerative Diseases, CHU de Bordeaux, Bordeaux, France. [44]Laboratory of Epidemiology and Population Science, National Institute on Aging, National Institutes of Health, Baltimore, MD, USA. [45]Department of Health Systems and Population Health, University of Washington, Seattle, WA, USA. [46]Institute of Molecular Medicine, McGovern Medical School, University of Texas Health Science Center at Houston, Houston, TX, USA. [47]These authors contributed equally: Adrienne Tin, Alison E. Fohner. [48]These authors jointly supervised this work: Sudha Seshadri, Myriam Fornage. ✉email: atin@umc.edu; afohner@uw.edu

