## [Peer Review File · Communications Biology]

Reviewers' comments:

Reviewer #1 (Remarks to the Author):

Summary:

Tin, Fohner and colleagues performed a study to identify plasma proteins associated with cognitive function. The authors first performed differential protein abundance analyses in two stages: discovery and replication. The authors next investigated the gene sets using the enrichment analysis on the differentially abundant proteins. The authors finally inferred the relationship between proteins on Alzheimer disease (AD) and cognitive function with bi-directional MR analysis. The authors used 3 cohorts (ARIC, CHS, FHS-Gen3) for the discovery stage of differential abundance analysis; for the replication stage, 4 cohorts (AGES, BLSA, MESA, Whitehall-II) using the SomaScan platform and 5 cohorts (3C, CARDIA, LBC-1936, LBC-1921, Rhineland) using the Olink platform were used. The authors performed three types of differential abundance analysis using different outcomes/participants: 1) General cognitive function (PC1 from all possible cognitive scores) with participants whose age is above or equal to 25 years old; 2) General cognitive function (PC1 from all possible cognitive scores) with participants above or equal to 65 years old; 3) Digit Symbol Substitution Test (DSST) cognitive scores alone with participants above or equal to 65 years old. The authors further explored protein-protein correlations, added eGFR as additional covariates for sensitivity check, and conducted gene set enrichment analyses. Finally, the authors inferred forward and reverse effects between these proteins and cognitive function or AD, highlighting NECTIN2 on AD, AD on CRP. Overall, this study meta-analyzed multiple large-scale plasma proteomics datasets (both participant level and protein level) with cognitive tests available. I do, however, have some comments before it gets published in Communications Biology.

Comments:

1. Study design: In Figure 1, the authors plotted the brief workflow for major analyses performed in this manuscript. However, to help the readers better understand the exact analyses given different combinations of the cohorts/proteins used for different outcome/age analyses, I think a detailed figure should be added either as a main figure or a supplementary figure. For example, 1049 proteins were used in the discovery of stage of all 3 cohorts (ARC, CHS, FHS-Gen3) when testing cognitive function (aged ≥ 25) & cognitive function (aged ≥ 65); 4709 proteins were used in the discovery of stage of only 2 cohorts (ARC, CHS) when testing cognitive function (aged ≥ 65) & DSST (aged ≥ 65).
2. As the authors used SomaScan platforms (1k, v4, v4.1), how are multiple aptamers corresponding to the same protein being processed? Do they account for independent proteins in the Bonferroni correction in the manuscript? If so, please clarify the concept of the aptamers and proteins in the corresponding section (results and methods).
3. Lines 173-177 and lines 181-186 are redundant, please correct the text redundancy.
4. Throughout the manuscript, the meta-analysis findings did not remove the $I^2 > 70$ or P -heterogeneity < 0.05 . If the authors are not going to update the findings, at least the authors should add this into the limitation section to explicitly note to the readers.
5. Line 189, EFNA4 protein was replicated using Olink platform despite its low correlation with SomaScan platform ($r=0.2$ per Supplementary Table 13). Could the authors comment on why this is happening?
6. For the pQTL datasets used in MR analysis section, the authors used a meta-analysis result from three studies (Sun et al 2018, Pietzner et al 2021, Ferkingstad et al 2021), did the authors check the I^2 for each pQTL used as IV in this study? If not, please add it as a limitation as some pQTLs may have high heterogeneity after meta-analyses.

7. For TwoSampleMR package used in the MR analyses, please clarify why did you use $r^2 < 0.01$ rather than the default 0.001 when clumping the data? This also may explain why SCG3 has 29 independent cis-pQTLs used in the MR analyses on cognitive function.
8. In all MR analyses, the authors used Egger intercept p-value and other robust methods for claiming no pleiotropy. This method, however, is not stringent and can lead to false positive findings, especially in the case of using AD risk as exposure. For example, in the manuscript, APOE region leads to significant associations, as none of the 10 significant associations remain significant after using SNPs outside the APOE region (Kunkle et al 2019). The similar findings were also observed in MR analysis using the Jansen et al 2019 as exposure. If the authors are not going to revise the MR analyses after removing the top-5 pleiotropic regions reported by Sun et al 2018, Pietzner et al 2021, Ferkingstad et al 2021, a major point of limitation needs to be added.
9. For genetic colocalization analysis, why did the authors only perform coloc with NECTIN2 on AD, but not other significant MR findings using proteins as exposure? For example, four proteins affecting cognitive function (PTK7, DNAJB12, SCG3, ITIH3).
10. Lines 258-260, did the authors identify such eQTL associations in the brain tissues from GTEx? If not, maybe the authors can also use the MetaBrain resource by N. de Klein et al 2023 (<https://www.nature.com/articles/s41588-023-01300-6>) to check.
11. Line 270, please report the exact PP.H4 from the coloc single-causal variant analysis, not just the < 0.05 .
12. Lines 574-578, please clarify the following questions regarding the coloc.susie analysis: How many variant-pairs were tested in the coloc.susie output? Please provide the reference genotype you used to derived the LD matrix from NECTIN2 and AD?
13. Line 579, which pair of PP.H4 from coloc.susie output did you use to make conclusions?

Reviewer #2 (Remarks to the Author):

The authors conducted an association study to explore the correlations between circulating proteins and general cognitive function in a large-scale sample. In the discovery phase, they identified a total of 246 proteins associated with cognitive function. Of these, 45 were replicated using the SomaScan platform and three were replicated using the Olink platform at Bonferroni-corrected significance. Enrichment analysis linked the proteins associated with general cognitive function to cell signaling pathways and synapse architecture. MR analysis implicated higher levels of NECTIN2, a protein mediating viral entry into neuronal cells, with higher AD risk. Levels of 20 other proteins were implicated as consequences of AD susceptibility ($p < 2.0E-4$), with the strongest effect observed for C-reactive protein. Overall, this is an interesting and valuable study. The methods used in this paper are appropriate, and the findings are interesting. I have several suggestions and concerns:

1. In the discovery phase, the authors identified a total of 246 proteins associated with cognitive function. However, only 45 were replicated using the SomaScan platform and three were replicated using the Olink platform. The authors need to clarify or discuss why the replication rate is so low. Considering the low replication rate, focusing on replicated proteins will provide more important information.
2. Another issue need to be justified is that the authors investigated circulating proteins. How these proteins contribute to general cognitive function ? Is it possible that these proteins exert their effect on general cognitive function through entering the brain ? Or these proteins have similar expression pattern in brain and blood ? These issues or possibilities needed to be discussed.
3. There were several large-scale genetic studies that investigated the associations between genetic variants and general cognitive function. Whether the genes encode the cognition-associated proteins (discovered in this study) showed associations with general cognitive function in previous genetic studies ?

4. MR analysis implicated higher levels of NECTIN2, a protein mediating viral entry into neuronal cells, with higher AD risk. Did any previous evidence show association between NECTIN2 and AD ? If there were studies that Knocked-down or knocked-out NECTIN2 in mice ? If so, did NECTIN2 knocked-out mice show abnormality in cognition ?

Reviewer #3 (Remarks to the Author):

The authors used populational cohorts to look for associations between protein levels using the Somalogic aptamer proteomics assay and cognitive function, and subsequently perform downstream enrichment and Mendelian randomization characterization of significant associations. Overall the paper is well structured with a good range of independent replication, although the reduced replication (despite large replication cohort design) is a slight concern. The ensuing downstream analyses also could benefit from a replication design since these data are available like for the primary analyses, and also benefit from replication/meta-analyses with already publicly available summary stats to increase robustness for the MR findings.

I have the following comments:

1. Are there other differences for other variables measured across the cohorts – those may also be important confounders.
2. The replication cohort, especially somascan ones, totals more than the discovery cohort which is slightly counterintuitive. It makes more sense to have more cohorts included as part of the discovery than in replication to maximise the benefit of this design – or was this due to data sharing challenges?
3. Given the previous large proteomic studies for dementia related phenotypes/disease risk, it's important to know how the results overlap with previous studies more systematically, and what proportions have been seen previously in related phenotypes.
4. ARIC and CHS used essentially the same assay but had different protein exclusion criteria, also some unqualified/unexplained part regarding the different normalization methods used and potential/examined impacts. Also ST4, C9 = 5068, not too sure what this is.
5. The PC from the cognitive scores: what variance does the other PCs explain, and would be good to know the correlations of the phenotypes for the traits used in the PC, since highly correlated ones would lead to first few PCs explaining most of the variance which is more suitable than if the cognitive domains were independent
6. Can authors comments on the heterogeneities & beta inconsistencies between the studies for overlapping proteins? There seem to be quite high I2 for a range of associations in ST7. Would be good to see how correlated the beta/test scores are across different cohorts to visualise consistencies globally too.
7. Can the authors provide rationale/reference to the choice 25 and 65 splits? So lower samples size of >65 vs >25 led to more significant findings, this is slightly counterintuitive from sample size point of view, simply excluding a load of proteins due to not being measured in one cohort is not ideal as those data can still provide useful information. Some clarifty here is needed to avoid misinterpreting the results as simply >65 have more proteins associated than >25.
8. Do the >25 and >65 results align well? If not, is there valid explanations/sensitivities done at different age cut-offs to check stability of the results?
9. What is the replication p threshold? It's a little confusing as to which Bonferroni threshold the text refers to.
10. The replication numbers, would be good to know the denominator/proportion the replicated (missing in table? and does the proportion of replication correspond to anticipated due to power.
11. The Olink replication rate is lower vs Somalogic, which is not unsurprising both intuitively and from other literature. It would be useful to refer to those studies.

12. The eGFR adjustments are very useful as a key confound in any age-related investigations, possibly along with blood cell counts/liver function(if available). It does however raise the issue that most of the findings here are possibly confounded by renal effects? Are the sets of protein that are affected by renal adjustment enriched for renal-cleared proteins or specific groups? Do any of the top findings change significantly or are the effects more spread out across the effect size/significance spectrum? Also are any effects reversed in direction as a result of eGFR adjustments? I feel there are more that can be done here rather than just state it needs looking at, since eGFR may be important to account for in future/other studies for cognitive function. Having an eGFR adjusted summary stat is also useful to the community.
13. It's not clear what was the background used for the enrichment analyses. The non-random selection of targets from all gene encoding background means the background should be accounted for or else one would get erroneous enrichments. The authors should at minimum account for the somascan background.
14. For the MR, is it just the discovery data used or the replication used? – any sensitivity analyses/replication done for the MR as was done for the primary analyses? Any that are robustly associated across assays?
15. The MR results need a replication cohort especially given in the absence of any benchmarked standards. Given the availability of protein instruments in other studies and in the replication cohorts, it would be useful to look at the robustness of the findings in the two-sample MR framework.
16. Since eGFR and renal function may be a mediator/driver between the instrument and the outcome, have the authors considered accounting for eGFR in the MR models? If not why?
17. The AD CRP effect is better looked at using larger datasets like UKB that have CRP in much larger sample sizes – do the authors replicate the CRP findings there?
18. The effects of APOE being pleiotropic is not something new. Are there any functional clusters of proteins that point to any indications of pathways/protein features driven by APOE?

Reviewer #1

1. Study design: In Figure 1, the authors plotted the brief workflow for major analyses performed in this manuscript. However, to help the readers better understand the exact analyses given different combinations of the cohorts/proteins used for different outcome/age analyses, I think a detailed figure should be added either as a main figure or a supplementary figure. For example, 1049 proteins were used in the discovery of stage of all 3 cohorts (ARC, CHS, FHS-Gen3) when testing cognitive function (aged ≥ 25) & cognitive function (aged ≥ 65); 4709 proteins were used in the discovery of stage of only 2 cohorts (ARC, CHS) when testing cognitive function (aged ≥ 65) & DSST (aged ≥ 65).

Response: Thank you for the ideas. We have revised Figure 1 as suggested. Please note the page and paragraph numbers in the responses below refer to the revised clean version of the manuscript.

2. As the authors used SomaScan platforms (1k, v4, v4.1), how are multiple aptamers corresponding to the same protein being processed? Do they account for independent proteins in the Bonferroni correction in the manuscript? If so, please clarify the concept of the aptamers and proteins in the corresponding section (results and methods).

Response: For multiple aptamers corresponding to the same proteins, we tested the values from each aptamers separately and used Bonferroni correction to account for all aptamers tested. In the manuscript, we refer to the measures from each aptamer as protein measures. In the Methods section, we clarified that the Bonferroni correction was applied for the total number of aptamers (protein measures) in each analysis, rather than the number of unique proteins (page 22, paragraph 1):

*“Given that some proteins were tagged by multiple aptamers, the values ascertained by each aptamer are referred as protein measures and analyzed separately.
“The statistical significance thresholds for discovery were Bonferroni corrected based on the number of protein measures in each analysis.”*

We have added the number of unique proteins in addition to the number of unique protein measures in **Table 1** and in the text of the Results section (page 6, paragraph 2):

*“A total of 1,049 protein measures annotated to 1,043 unique proteins were tested for general cognitive function among participants aged ≥ 25 .
“In the analysis of 4,709 protein measures (annotated to 4,506 proteins) from the two discovery cohorts with participants aged ≥ 65 , we identified ...”*

3. Lines 173-177 and lines 181-186 are redundant, please correct the text redundancy.

Response: Thank you for the careful reading of our manuscript. We have removed the redundant text.

4. Throughout the manuscript, the meta-analysis findings did not remove the $I^2 > 70$ or P -heterogeneity < 0.05 . If the authors are not going to update the findings, at least the authors should add this into the limitation section to explicitly note to the readers.

Response: We chose to report all protein measures reaching statistical significance in the discovery analysis, along with their corresponding I^2 , p -value of heterogeneity, and effect direction (**Supplementary Table 9**). We have not filtered our results for several reasons:

- 1) Providing complete information may be important for future studies, which may independently evaluate our findings.
- 2) Associated proteins that showed statistical evidence of between-study heterogeneity ($I^2 > 70$ or p -value for heterogeneity < 0.05) generally showed consistent direction of effects among discovery cohorts. For example, with general cognitive function among those aged ≥ 65 , 153 of the 211 significant proteins had $I^2 > 70\%$ or p -value for heterogeneity < 0.05 , and 150 of these 153 proteins had consistent effect direction between the discovery cohorts. This suggests the associations were supported by most studies although the effect size of the association varied between studies. These variations may relate to differences in settings, methods and populations between studies or be due to differences in genetic or non-genetic influences on protein levels.
- 3) Lastly, we believe that detection of heterogeneity may be important and needs to be reported because heterogeneity may be a reflection of the complex pattern of correlations among proteins. For example, a heterogeneous association for a particular protein may reflect a consistent association with another correlated protein that may vary between studies. Statistical heterogeneity may offer a window to the complexity in the data and should therefore not be discarded. We have noted this heterogeneity in the Discussion (page 18, paragraph 3).

“Among the protein measures that were significant in the discovery analysis, some had considerable heterogeneity, which may partly reflect different protein levels between middle age and older age as previously reported.”

5. Line 189, EFNA4 protein was replicated using Olink platform despite its low correlation with SomaScan platform ($r=0.2$ per Supplementary Table 13). Could the authors comment on why this is happening?

Response: Multiple factors could affect the correlation between measures from SomaScan and Olink, including binding affinity of the reagent (aptamer vs. antibody), glycosylation of the protein, and limit of detection of the assay.¹ The proprietary nature and the lack of full technical details of the two assays have limited the ability of the scientific community to investigate factors leading to the low correlation in some protein measures between the two platforms. As reported from a previous study, measures from the two platforms could provide complementary information on the association

between a protein and a trait.¹ Therefore, we conducted the replication study using Olink as an exploratory analysis. We have added the factors that may affect the correlation between protein measures from the two platforms to the manuscript (page 8, paragraph 2):

“It is known that multiple factors could affect the correlation of protein measures quantified using the two platforms, including binding affinity of the reagent (aptamer vs. antibody), glycosylation of the protein, and limit of detection of the assay.”

6. For the pQTL datasets used in MR analysis section, the authors used a meta-analysis result from three studies (Sun et al 2018, Pietzner et al 2021, Ferkingstad et al 2021), did the authors check the I^2 for each pQTL used as IV in this study? If not, please add it as a limitation as some pQTLs may have high heterogeneity after meta-analyses.

Response: Thank you for raising this important point. Overall, the heterogeneity was modest. Among the 3 proxy SNPs of the novel significant proteins (NECTIN2, DNAJB12, PTK7) in the forward MR analysis, the I^2 statistics were 39.1, 8.2, 85.7, respectively, and, importantly, all had consistent effect directions in the pQTL studies. Among the over 100 SNPs that were used in the reverse MR analysis, the median I^2 were 0 (1st and 3rd quartiles: 0, 35.4) for general cognitive function and 0 (1st and 3rd quartiles: 0, 16.4) for AD. These statistics are reported in **Supplementary Tables 26 and 30** and their respective sections in Results (page 10, paragraph 3; page 11, paragraph 2; page 12, paragraph 1; page 13, paragraph 2).

7. For TwoSampleMR package used in the MR analyses, please clarify why did you use $r^2 < 0.01$ rather than the default 0.001 when clumping the data? This also may explain why SCG3 has 29 independent cis-pQTLs used in the MR analyses on cognitive function.

Response: $r^2 < 0.01$ was the default r^2 threshold for LD clumping used by TwoSampleMR until the recent change to $r^2 < 0.001$. Given the comment from the reviewer, we have rerun the MR analysis using an r^2 threshold of 0.001. Many of the results we reported remain significant: DNAJB12 and PTK7 for the causal effect of proteins on general cognitive function, SLITRK3 for the causal effect of general cognitive function on protein, NECTIN2 for the causal effect of protein on AD susceptibility. The previously reported causal effect of SVEP1 on AD susceptibility remained replicated. On the causal effect of AD susceptibility on proteins, CRP, CTSZ, C1RL, and CERT1 remained significant. We have revised the manuscript to report the results using $r^2 < 0.001$. (Results section, pages 10 to 13; **Table 2, Supplementary Tables 25 to 39**).

8. In all MR analyses, the authors used Egger intercept p-value and other robust methods for claiming no pleiotropy. This method, however, is not stringent and can lead to false positive findings, especially in the case of using AD risk as exposure. For example, in the manuscript, APOE region leads to significant associations, as none of the 10 significant associations remain significant after using SNPs outside the APOE region (Kunkle et al 2019).

The similar findings were also observed in MR analysis using the Jansen et al 2019 as exposure. If the authors are not going to revise the MR analyses after removing the top-5 pleiotropic regions reported by Sun et al 2018, Pietzner et al 2021, Ferkingstad et al 2021, a major point of limitation needs to be added.

Response: We would like to clarify that we did not claim “no pleiotropy”. In MR, pleiotropy refers to the violation of the exclusion restriction assumption, which assumes that the proxy SNPs of the exposure influence the outcome only through the exposure. If a proxy SNP of the exposure influences the outcome through other pathways, this SNP is not a valid proxy of the exposure. The methods that are robust to pleiotropy of proxy SNPs can provide valid causal estimates even if some proxy SNPs are not valid proxies of the exposure. Indeed, the weighted mode method can provide consistent causal effect estimates even if the majority of the proxy SNPs are not valid proxies of the exposure. Therefore, our interpretation of the significant results from methods that are robust to pleiotropy is: the exposure is likely a causal factor on the outcome even in the presence of potential pleiotropy of some proxy SNPs. We have clarified this interpretation in the Methods section (page 26, paragraph 1):

“To evaluate the effect of an exposure in the presence of potential pleiotropic effect of the genetic proxies, we used methods robust to pleiotropy (Egger regression, weighted median, weighted mode) as secondary methods. These methods can provide valid causal estimates even when some proxy SNPs of the exposure may influence the outcome through pathways outside of the exposure and thus are not valid proxies of the exposure due to their pleiotropic effects.”

For the MR analysis using < 3 proxy SNPs, there were no established methods for assessing potential pleiotropy. We included this as a limitation (page 19, paragraph 1):

Regarding the significant causal effects of AD susceptibility from the APOE region on proteins, if the significant proteins were highly correlated, then the significant results of some proteins might be driven by other proteins. However, the significant proteins were only modestly correlated: abs(Pearson correlation of log2-transformed values), using Kunkle et al. AD summary statistics: median (1st, 3rd quartile): 0.13 (0.08, 0.18); using Jansen et al. AD summary statistics: median (1st, 3rd quartile): 0.17 (0.11, 0.23) (**Supplementary Table 37**).

Regarding the pleiotropic regions in the 3 pQTL studies, these are regions that contain SNPs that were associated with more than one protein in *cis* or *trans*. We have limited the proxy SNPs of a protein to be *cis*-SNP to reduce potential pleiotropic effect from *trans* proxy SNPs. In addition, we used Steiger filtering to exclude proxy SNPs that were associated with the outcome more strongly than the exposure to reduce the potential for reverse causation.

9. For genetic colocalization analysis, why did the authors only perform coloc with NECTIN2

on AD, but not other significant MR findings using proteins as exposure? For example, four proteins affecting cognitive function (PTK7, DNAJB12, SCG3, ITIH3).

Response: Thank you for this idea. We performed colocalization analysis for DNAJB12 and PTK7, the two proteins that had significant effect on general cognitive function using $r^2 < 0.001$ for LD clumping. The posterior probability for H4 was 86.3% for DNAJB12 and 68.3% for PTK7. The SuSiE method did not identify any 95% credible set from the summary statistics of general cognitive function in these two regions and thus did not produce posterior probability estimates. We have added this analysis in Methods (page 27, paragraph 1), Results (page 11, paragraph 1, **Supplementary Table 28**) and Discussion (page 15, paragraph 1)

10. Lines 258-260, did the authors identify such eQTL associations in the brain tissues from GTEx? If not, maybe the authors can also use the MetaBrain resource by N. de Klein et al 2023 (<https://www.nature.com/articles/s41588-023-01300-6>) to check.

Response: Thank you for this idea. We have queried the associations of the pQTLs of DNAJB12, PTK3, and NECTIN2 in the MetaBrain results (<https://www.metabrain.nl/cis-eqtls.html>). None of the pQTLs were associated with the expression of the target proteins at FDR < 0.05. We have added this information in the Results section (page 11, paragraph 3; page 12, paragraph 1).

11. Line 270, please report the exact PP.H4 from the **coloc** single-causal variant analysis, not just the < 0.05.

Response: As suggested, we revised the Results section to report exact posterior probability for H4 (**Supplementary Table 28**, page 11, paragraph 1; page 12, paragraph 2):

*“Colocalization analyses for these two proteins provided support for a single shared causal variant for DNAJB12 (posterior probability [PP] for H4 = 0.86, **Supplementary Table 28, Supplementary Figure 12**) and weak support for a shared causal variant for PTK7 (PP H4 = 0.68, **Supplementary Figure 13**).”*

*“Colocalization analyses within the 500kb region on both sides of the NECTIN2 promoter, which included the APOE gene, did not support a single shared causal variant underlying NECTIN2 protein levels and AD susceptibility (PP of H4: 2.0E-11, **Supplementary Table 28**).”*

12. Lines 574-578, please clarify the following questions regarding the **coloc.susie** analysis: How many variant-pairs were tested in the coloc.susie output? Please provide the reference genotype you used to derive the LD matrix from NECTIN2 and AD?

Response: The reference genome in the coloc analysis was the European ancestry in 1000 Genomes. We have clarified this in the Methods section (page 27, paragraph 2). We also added the information of the posterior probabilities of each variant pair from the coloc.susie analysis (**Supplementary Table 34**).

13. Line 579, which pair of PP.H4 from **coloc.susie** output did you use to make conclusions?

Response: For the coloc.susie results, we used the posterior probabilities of all pairs of variants. We clarified this in the Methods section (page 27, paragraph 2).

Reviewer #2 (Remarks to the Author):

Overall, this is an interesting and valuable study. The methods used in this paper are appropriate, and the findings are interesting.

Response: We thank the reviewer for the positive feedback. Please note that the page and paragraph numbers in the responses below refer to the revised clean version of the manuscript.

1. In the discovery phase, the authors identified a total of 246 proteins associated with cognitive function. However, only 45 were replicated using the SomaScan platform and three were replicated using the Olink platform. The authors need to clarify or discuss why the replication rate is so low. Considering the low replication rate, focusing on replicated proteins will provide more important information.

Response: Thank you for this important question. A driving factor in our replication is the limited sample size for replication. Based on our post-hoc power analysis, given the replication sample size from the SomaScan platform and the median effect size among the significant associations in the discovery study, the power for replication among participants aged ≥ 65 for general cognitive function and DSST were only 0.61 and 0.67, respectively (reported on page 7, paragraph 3). On the Olink platform, the power for replication were 0.18 to 0.29 for the 3 analyses (general cognitive function among aged ≥ 25 , general cognitive function among aged ≥ 65 , and DSST among aged ≥ 65 , reported on page 8, paragraph 2). In addition to sample size, SomaScan and Olink may not measure the same aspect of a protein's levels.¹ Therefore, we consider the replication analysis using Olink as exploratory.

The significance thresholds of our primary replication using SomaScan were Bonferroni-corrected, which is conservative considering the correlation between proteins. Consequently, our primary replication results were very robust. We also reported replication based on $FDR < 0.05$. Using data from SomaScan, the replication rates were 12.5% to 28.9% based on Bonferroni-corrected threshold and 39.9% to 65.8% based on $FDR < 0.05$. This information was reported in Methods (page 22, paragraph 2) and Results (page 8, paragraph 1 and **Table 1**).

We have added limited replication sample size and heterogeneity in protein platform in the Discussion section (page 18, paragraph 3).

2. Another issue need to be justified is that the authors investigated circulating proteins. How these proteins contribute to general cognitive function? Is it possible that these proteins exert their effect on general cognitive function through entering the brain? Or these proteins have similar expression pattern in brain and blood? These issues or possibilities needed to be discussed.

Response: Thank you for these insightful questions. Some proteins enter the bloodstream by purposeful secretion to orchestrate biological processes (e.g. cytokines, chemokines, adipokines, hormones, growth factors) while other proteins enter blood through leakage from cell damage and cell death. Both secreted and leakage proteins can inform health status and disease risk.² Vascular dysfunction has long been hypothesized as an important component of AD pathophysiology.³ Systemic infection and inflammation could increase blood-brain barrier permeability and provide the opportunities for circulating proteins to affect brain function.^{4, 5} It is also possible that circulating proteins reflect brain health that affects cognitive functions. We have added these points in the Introduction section (page 5, paragraph 1).

3. There were several large-scale genetic studies that investigated the associations between genetic variants and general cognitive function. Whether the genes encode the cognition-associated proteins (discovered in this study) showed associations with general cognitive function in previous genetic studies?

Response: The largest GWAS on general cognitive function to date is by Davies et al. 2018. We used the results from Davies et al. for the Mendelian randomization analysis on the relationship between protein and general cognitive function. For the 246 proteins that were significantly associated cognitive function in the analyses among aged ≥ 25 or aged ≥ 65 in our discovery study, we interrogated the regions 500kb on both sides of the promoter of the *encoding* gene in Davies et al. and found 38 proteins had one or more SNPs with p-value $< 5e-8$. This interrogation is now reported in Methods (page 23, paragraph 2), Results (page 7, paragraph 2) and **Supplementary Table 13**.

4. MR analysis implicated higher levels of NECTIN2, a protein mediating viral entry into neuronal cells, with higher AD risk. Did any previous evidence show association between NECTIN2 and AD? If there were studies that Knocked-down or knocked-out NECTIN2 in mice? If so, did NECTIN2 knocked-out mice show abnormality in cognition?

Response: Thank you for these insightful questions. Associations between variants in *NECTIN2*, aka *PVRL2*, and AD independent of the APOE e4 variants have been reported.^{6, 7} In the revised manuscript, these publications are cited in page 16, paragraph 3.

NECTIN2 knockout mice were reported to have degeneration of astrocytic perivascular end foot processes and neurons in the cerebral cortex, although cognition-specific

phenotypes were not reported.⁸ This information is added to the Discussion (page 16, paragraph 3).

Reviewer #3 (Remarks to the Author):

Overall the paper is well structured with a good range of independent replication.

Response: We thank the reviewers for the positive feedback. Please note the page and paragraph numbers in the responses below refer to the revised clean version of the manuscript.

1. Are there other differences for other variables measured across the cohorts – those may also be important confounders.

Response: Regarding the population characteristics of the 3 discovery cohorts, we have added clinical characteristics that are considered dementia risk factors (prevalent diabetes and hypertension, smoking status, and drinking status) in **Supplementary Table 1**. The proportion of participants with prevalent diabetes ranged from 0.56% in FHS Gen 3 to 26.6% in ARIC, and those with hypertension ranged from 3.54% in FHS Gen 3 to 75.19% in CHS. We chose not to control for these clinical characteristics in our analysis given the goal of our cross-sectional study was to identify circulating biomarkers of general cognitive function including those affected by dementia risk factors. We chose to use Mendelian randomization analysis to address the potential causal relationship between the identified proteins and general cognitive function.

2. The replication cohort, especially somascan ones, totals more than the discovery cohort which is slightly counterintuitive. It makes more sense to have more cohorts included as part of the discovery than in replication to maximise the benefit of this design – or was this due to data sharing challenges?

Response: It is indeed due to data sharing and timing challenges. The project was executed on a fixed timeline. Cohorts included in the discovery were those that were able to share a full set of association results in a timely manner. Some cohorts could only provide limited results (replication results); others did not have proteomics data during the discovery phase of the project and obtained proteomics data later. These cohorts were thus included in the replication analyses.

3. Given the previous large proteomic studies for dementia related phenotypes/disease risk, it's important to know how the results overlap with previous studies more systematically, and what proportions have been seen previously in related phenotypes.

Response: Thank you for this suggestion. Of the 220 protein measures that were significantly associated with general cognitive function in the discovery analyses among

participants aged ≥ 25 or 65 , 20 were associated with incident dementia in the ARIC study, one of the discovery cohort, 67 were associated with cognitive decline, and 21 were associated with general cognitive ability. These results are reported in page 7, paragraph 2, and **Supplementary Table 12**.

4. ARIC and CHS used essentially the same assay but had different protein exclusion criteria, also some unqualified/unexplained part regarding the different normalization methods used and potential/examined impacts. Also ST4, C9 = 5068, not too sure what this is.

Response: Thank you for carefully reading the paper. Cell C9 should be the normalization method used by AGES. This has been corrected. The field of proteomics has yet to develop a common standard for filtering out low quality proteins. Each cohort analyzed the proteins based on their quality control criteria. We include only the proteins that were present in all cohorts. So our method was conservative. Our results are likely to be robust given that the protein measures passed the quality control measures of all cohorts.

5. The PC from the cognitive scores: what variance does the other PCs explain, and would be good to know the correlations of the phenotypes for the traits used in the PC, since highly correlated ones would lead to first few PCs explaining most of the variance which is more suitable than if the cognitive domains were independent.

Response: Thank you for the insight. The variances explained by PC1 were 0.47 to 0.67 among the discovery cohorts and 0.47 to 0.69 among the replication cohorts (**Supplementary Table 7**). The correlation between the cognitive scores in each discovery cohorts were moderate (range: 0.11 to 0.54, **Supplementary Table 5**). These low to moderate correlations between cognitive scores suggest the different scores may indeed reflect different domains of cognition.

6. Can authors comments on the heterogeneities & beta inconsistencies between the studies for overlapping proteins? There seem to be quite high I² for a range of associations in ST7. Would be good to see how correlated the beta/test scores are across different cohorts to visualise consistencies globally too.

Response: In the discovery study, beta directions were highly consistent among the cohorts with participants aged ≥ 65 ($> 90\%$ of the significant proteins). For the significant proteins, betas between ARIC and CHS were highly correlated (0.86 to 0.88) across the 3 analyses. In contrast, the betas of ARIC and CHS had low correlation with those of FHS Gen3 (0.002 between ARIC and FHS Gen3, 0.06 between CHS and FHS Gen3). The sources of heterogeneity may include the version of the protein assay, and the age of the study population. These results are also consistent with previous reports on the differences in protein levels or associations between middle age and older age.⁹

¹⁰ We have added the correlations of betas in the Results (page 7, paragraph 1 and **Supplementary Table 11**).

7. Can the authors provide rationale/reference to the choice 25 and 65 splits? So lower samples size of >65 vs >25 led to more significant findings, this is slightly counterintuitive from sample size point of view, simply excluding a load of proteins due to not being measured in one cohort is not ideal as those data can still provide useful information. Some clarify here is needed to avoid misinterpreting the results as simply >65 have more proteins associated than >25.

Response: Having separate analysis for participants aged ≥ 65 was motivated by previous research that showed substantial differences in protein levels between middle age and older age.⁹ This is further supported by another study, which reported that some proteins with significant associations among older individuals were not replicated among younger individuals.¹⁰ More significant proteins were reported among those aged ≥ 65 likely due to the number of available proteins among cohorts. FHS Gen3 was the only cohort with participants aged < 65 and used an assay with ~ 1000 proteins. Both ARIC and CHS contributed results from participants aged ≥ 65 and used an assay with $>4,000$ proteins. If the analysis among participants ≥ 25 did not require the protein to be present in all studies, then many significant results among aged ≥ 25 would be exclusively driven by participants aged ≥ 65 from ARIC and CHS. Thus, presenting these results from ARIC and CHS as significant across all age groups would be a misrepresentation. We have added more clarification as to this rationale in the Methods section (page 21, paragraph 2):

“Given that the meta-analysis of general cognitive function among those aged ≥ 25 only had one cohort with participants aged < 65 and the meta-analysis among those age ≥ 65 only had two cohorts, we required that each meta-analysis included only protein measures that were present in all cohorts.”

8. Do the >25 and >65 results align well? If not, is there valid explanations/sensitivities done at different age cut-offs to check stability of the results?

Response: Of the 79 protein measures that were significantly associated with general cognitive function among all participants aged ≥ 25 , 70 were also significantly associated with general cognitive function among participants aged ≥ 65 (Results section, page 6, paragraph 2). We would like to note that participants aged ≥ 65 were included in the analysis among aged ≥ 25 .

9. What is the replication p threshold? It's a little confusing as to which Bonferroni threshold the text refers to.

Response: Thank you for pointing this out. For the Bonferroni-corrected p-value thresholds for replication, we corrected for the number of protein measures that were selected for replication and also available from the replication cohorts in each analysis. We have added a table with all the Bonferroni-corrected p-value thresholds and hope that this is now clear (**Supplementary Table 8**, Results, page 7, paragraph 3).

10. The replication numbers, would be good to know the denominator/proportion the replicated (missing in table? and does the proportion of replication correspond to anticipated due to power.

Response: We added the number of proteins tested in replication in the **Supplementary Tables 15 to 18**. It is not straightforward to compare the proportion of replication with power. Each protein has its own post-hoc power for replication which depends on its effect size. We reported post-hoc power to provide a sense for the probability of replication given the replication sample size and assuming that the associations are homogeneous across the populations. As has been shown, the association of a circulating protein with cognitive function can be highly affected by age.¹⁰ In addition, with the replication using Olink data, the two assays may measure different aspects of protein levels.¹¹ Hence we considered the replication using Olink data as exploratory. Our replication results highlight the current challenges in the study of proteomics, such as heterogeneity in assay platforms and the need to further characterize factors that affect protein associations. We added these points in the Discussion (page 18, paragraph 3).

11. The Olink replication rate is lower vs Somalogic, which is not unsurprising both intuitively and from other literature. It would be useful to refer to those studies.

Response: A previous study that analyzed the cross-sectional association between protein and cognitive function using Olink data reported heterogeneity of protein association between cohorts of similar age.¹⁰ In our study, several factors might have contributed to the lower replication rate using Olink data compared with SomaScan. These factors include smaller replication sample size using Olink data than SomaScan data and that the two protein platforms may quantify different aspects of protein levels.¹ We have expanded this point in the Discussion (page 18, paragraph 3).

12. The eGFR adjustments are very useful as a key confound in any age-related investigations, possibly along with blood cell counts/liver function(if available). It does however raise the issue that most of the findings here are possibly confounded by renal effects? Are the sets of protein that are affected by renal adjustment enriched for renal-cleared proteins or specific groups? Do any of the top findings change significantly or are the effects more spread out across the effect size/significance spectrum? Also are any effects reversed in direction as a result of eGFR adjustments? I feel there are more that can be done here rather than just state it needs looking at, since eGFR may be important to account for in future/other studies for cognitive function. Having an eGFR adjusted summary stat is also useful to the community.

Response: Thank you for raising these interesting points. We will make available the summary statistics adjusting for eGFR in the same way as we made the discovery summary statistics available.

We considered whether kidney function might be a potential confounder or mediator in the relationship between protein and cognitive function in a causal sense. Linear regression cannot separate statistical correlation from confounding or mediation. Being a potential confounding or mediating factor implies that the factor is likely a causal factor of the cognitive function. While there are reports of the association of kidney function with dementia and AD,^{12, 13} two large-scale MR studies reported no evidence supporting a causal effect of kidney function on dementia or AD.^{14, 15} A large-scale proteomic study using SomaScan reported most proteins are markers of kidney function without any causal relationship.¹⁶ Only one protein (LMAN2) might be a causal factor for kidney function with support from MR analysis. Additionally, we have conducted MR analysis of kidney function on general cognitive function and did not find support for kidney function as a causal factor of general cognitive function (IVW MRE beta=-0.004, p=9.71E-1, Results, page 9 paragraph 2, Supplementary Methods and Results). Kidney function is correlated with protein levels due to glomerular filtration. Therefore, the change in association between a protein and general cognitive function after adjusting for eGFR is likely induced by statistical correlation rather than confounding and mediation in the causal sense. If the purpose of a study were to use circulating protein measures to obtain a precise prediction of cognitive function, then the statistical correlation between eGFR and protein measures would be more relevant to our analysis.¹⁷ These points are added to the Discussion (page 17, paragraph 2):

“Our sensitivity analysis adjusting for kidney function showed that the association of some protein measures were attenuated. However, two large-scale MR studies reported no evidence supporting the causal effect of kidney function on dementia or AD.^{53, 54} Our MR analysis also did not support kidney function as a potential causal factor of general cognitive function. A large-scale proteomic study using the SomaScan platform only implicated one protein out of almost 5000 as a potential causal factor of kidney function and suggested that most proteins are likely markers of kidney function.⁵⁵ Therefore the attenuation of the association between protein measures and cognitive function after adjusting for eGFR was likely due to statistical correlation between eGFR and protein measures rather than causal relationships between kidney function and cognitive function.”

13. It's not clear what was the background used for the enrichment analyses. The non-random selection of targets from all gene encoding background means the background should be accounted for or else one would get erroneous enrichments. The authors should at minimum account for the somascan background.

Response: For the enrichment analysis, the background was the encoding genes of all protein measures tested in the discovery analysis, as recommended by the Reviewer. We have further clarified this point in the Methods section (page 24, paragraph 1).

14. For the MR, is it just the discovery data used or the replication used? – any sensitivity

analyses/replication done for the MR as was done for the primary analyses? Any that are robustly associated across assays?

Response: All data used in the two-sample MR were independent of the data used in the discovery analysis. Please see the response to the next comment regarding the replication of MR results. Regarding MR across assays, our two-sample MR analysis used pQTL data generated from SomaScan data, given that our discovery study used data from SomaScan, and SomaScan and Olink may quantify different aspects of proteins levels.¹ For the proteins whose associations with cognitive function were replicated using Olink data, a possible follow-up would be an MR analysis using Olink data. Unfortunately, the publicly available pQTL dataset using Olink data¹⁸ does not include the proteins that were replicated in our analysis using Olink data.

15. The MR results need a replication cohort especially given in the absence of any benchmarked standards. Given the availability of protein instruments in other studies and in the replication cohorts, it would be useful to look at the robustness of the findings in the two-sample MR framework.

Response: Our MR analysis followed recommendations and standard practices from the Guidelines for Performing Mendelian Randomization Investigations.¹⁹ A replication of the MR findings requires separate independent datasets for both exposure and outcome. Kunkle et al. 2019 is the only large-scale dataset for AD among European ancestry with clinical AD as the phenotype. Jansen et al. 2019 included AD by proxy as cases and its data included those from Kunkle et al. Bellenguez et al. 2022 excluded the APOE region from its publicly available summary statistics. Therefore, these summary statistics are not adequate for replicating the results of the two-sample MR using the Kunkle et al. dataset. With general cognitive function, Davies et al. 2018 is also the only large-scale GWAS dataset. Thus, we were limited by available datasets for these analyses. We hope that as the number of GWAS studies on AD and cognitive function increases, independent replication would be feasible.

16. Since eGFR and renal function may be a mediator/driver between the instrument and the outcome, have the authors considered accounting for eGFR in the MR models? If not why?

Response: As explained in the response to comment #12, for kidney function to be a potential confounder or mediator in the relationship between protein and cognitive function, kidney function would be a causal factor of cognitive function. However, large-scale MR studies reported no evidence supporting the causal effect of kidney function on dementia or AD.^{14, 15} In response to this comment, we have performed MR analysis of kidney function on general cognitive function and did not find significant causal effect (IVW MRE beta=0.004, p-value=9.71E-1, Results, page 9 paragraph 2, Supplementary Methods and Results). In the presence of all the evidence rejecting the potential causal role of kidney function on AD, dementia, and cognitive function, there is

a lack of rationale to formulate a causal hypothesis for including eGFR in the MR analysis.

17. The AD CRP effect is better looked at using larger datasets like UKB that have CRP in much larger sample sizes – do the authors replicate the CRP findings there?

Response: We agree that UKB is a valuable resource. We searched the GWAS catalog and found 3 GWAS of CRP with UKB as one of the contributing cohorts. Similarly to our results, the 2 risk variants of APOE ϵ 4 (rs429358 C allele and rs7412 C allele) were significantly associated with lower CRP levels. This additional evidence has been added in Methods (page 28, paragraph 2), Results (page 14, paragraph 2, **Supplementary Table 41**).

18. The effects of APOE being pleiotropic is not something new. Are there any functional clusters of proteins that point to any indications of pathways/protein features driven by APOE?

Response: In addition to CRP, some significant results are consistent with pathways known to be dysregulated in AD. For example, dysregulation in ubiquitin signaling and lysosomal function has been known in AD.^{20, 21} Two lysosomal cysteine proteinase (CTSA and CTSZ) and an ubiquitin conjugating enzyme (UBE2G2) were found to be affected by AD susceptibility from the APOE region. We added this information to the Discussion (page 17, paragraph 1).

References

1. Pietzner M, *et al.* Synergistic insights into human health from aptamer- and antibody-based proteomic profiling. *Nature Communications* **12**, 6822 (2021).
2. Pontén F, Schwenk JM, Asplund A, Edqvist P-HD. The Human Protein Atlas as a proteomic resource for biomarker discovery. *Journal of Internal Medicine* **270**, 428-446 (2011).
3. Sweeney MD, *et al.* Vascular dysfunction-The disregarded partner of Alzheimer's disease. *Alzheimer's & dementia : the journal of the Alzheimer's Association* **15**, 158-167 (2019).
4. Galea I. The blood–brain barrier in systemic infection and inflammation. *Cellular & molecular immunology* **18**, 2489-2501 (2021).
5. Sweeney MD, Sagare AP, Zlokovic BV. Blood-brain barrier breakdown in Alzheimer disease and other neurodegenerative disorders. *Nat Rev Neurol* **14**, 133-150 (2018).

6. Zhou X, *et al.* Non-coding variability at the APOE locus contributes to the Alzheimer's risk. *Nat Commun* **10**, 3310 (2019).
7. Logue MW, *et al.* A comprehensive genetic association study of Alzheimer disease in African Americans. *Arch Neurol* **68**, 1569-1579 (2011).
8. Miyata M, *et al.* Localization of nectin-2 δ at perivascular astrocytic endfoot processes and degeneration of astrocytes and neurons in nectin-2 knockout mouse brain. *Brain research* **1649**, 90-101 (2016).
9. Lehallier B, *et al.* Undulating changes in human plasma proteome profiles across the lifespan. *Nature Medicine* **25**, 1843-1850 (2019).
10. Harris SE, *et al.* Neurology-related protein biomarkers are associated with cognitive ability and brain volume in older age. *Nature Communications* **11**, 800 (2020).
11. Li Y, *et al.* Genome-wide studies reveal factors associated with circulating uromodulin and its relationships to complex diseases. *JCI Insight* **7**, (2022).
12. Xu H, *et al.* Kidney Function, Kidney Function Decline, and the Risk of Dementia in Older Adults. *Neurology* **96**, e2956 (2021).
13. Scheppach JB, *et al.* Albuminuria and Estimated GFR as Risk Factors for Dementia in Midlife and Older Age: Findings From the ARIC Study. *Am J Kidney Dis* **76**, 775-783 (2020).
14. Kjaergaard AD, Ellervik C, Witte DR, Nordestgaard BG, Frikke-Schmidt R, Bojesen SE. Kidney function and risk of dementia: Observational study, meta-analysis, and two-sample mendelian randomization study. *Eur J Epidemiol* **37**, 1273-1284 (2022).
15. Liu X, Ou YN, Ma YH, Huang LY, Zhang W, Tan L. Renal function and neurodegenerative diseases : a two-sample Mendelian randomization study. *Neurological research* **45**, 456-464 (2023).
16. Grams ME, *et al.* Proteins Associated with Risk of Kidney Function Decline in the General Population. *Journal of the American Society of Nephrology* **32**, (2021).
17. Dittrich A, *et al.* Association of Chronic Kidney Disease With Plasma NfL and Other Biomarkers of Neurodegeneration: The H70 Birth Cohort Study in Gothenburg. *Neurology* **101**, e277-e288 (2023).
18. Folkersen L, *et al.* Genomic and drug target evaluation of 90 cardiovascular proteins in 30,931 individuals. *Nature metabolism* **2**, 1135-1148 (2020).

19. Burgess S, *et al.* Guidelines for performing Mendelian randomization investigations [version 2; peer review: 2 approved]. *Wellcome Open Research* **4**, (2020).
20. Schmidt MF, Gan ZY, Komander D, Dewson G. Ubiquitin signalling in neurodegeneration: mechanisms and therapeutic opportunities. *Cell Death & Differentiation* **28**, 570-590 (2021).
21. Gowrishankar S, *et al.* Massive accumulation of luminal protease-deficient axonal lysosomes at Alzheimer's disease amyloid plaques. *Proc Natl Acad Sci U S A* **112**, E3699-3708 (2015).

REVIEWERS' COMMENTS:

Reviewer #1 (Remarks to the Author):

Summary:

Tin, Fohner and colleagues have addressed most of my comments from the previous round. I do have four minor comments for this round that I would like the authors to address:

Comments:

1. To address the low replication rate issue, you calculated the power to estimate whether the replication would be successful or not. However, I don't think current version provides enough details in the Methods section. Lines 515-517, the information only mentioned effect size per discovery and sample size per replication, but no statistical models or packages were documented. It would be helpful to elaborate this analysis in the Methods section.
2. From lines 581-583, you mentioned that the genetic proxies for protein measure were essentially "cis-pQTLs". I would suggest that you explicitly using this term. This would help clarify the IVs used here, given the three pQTL studies identified both cis and trans pQTLs.
3. Given the authors did not identify the cis-genes with the brain tissue (either MetaBrain or GTEx), I think it is worth discussing that tissue types, i.e. blood and brain, may pinpoint different causal genes/proteins to different neurological diseases/traits. In the introduction section, the authors justified the proteins from blood can be used to study brain diseases, but I think this new discussion/interpretation would be helpful to be added in the manuscript in the Discussion section.
4. For the pQTL datasets used in MR analysis section in line 562, the authors used a meta-analysis result from three studies (Sun et al 2018, Pietzner et al 2021, Ferkingstad et al 2021). Is the "fixed-effect meta-analysis" implemented in METAL or something else? I am asking as I found in lines 519-520, the authors mentioned the tool they used to perform meta-analysis on the differential protein abundance analysis, but not mentioning the tool for the pQTL used for MR. Also, the authors need to provide more details on how they harmonize different human genome reference builds of these three studies given that Sun et al 2018 used hg19, Pietzner et al 2021 used hg19, but Ferkingstad et al 2021 used hg38. Did you liftover hg38 to hg19 before meta-analyzing all three studies?

Reviewer #2 (Remarks to the Author):

The authors have addressed my concerns adequately, I have no further comments.

Reviewer #3 (Remarks to the Author):

Thank you to the authors for their responses. I have no major outstanding issues. The only comment I would like, given the utility of downstream applications of the results, is to make bulk download of the summary statistics accessible and easy through command line.

Reviewer #1 (Remarks to the Author):

Summary:

Tin, Fohner and colleagues have addressed most of my comments from the previous round. I do have four minor comments for this round that I would like the authors to address:

Comments:

1. To address the low replication rate issue, you calculated the power to estimate whether the replication would be successful or not. However, I don't think current version provides enough details in the Methods section. Lines 515-517, the information only mentioned effect size per discovery and sample size per replication, but no statistical models or packages were documented. It would be helpful to elaborate this analysis in the Methods section.

Response: We calculated the post-hoc power using the R function `pwr.f2.test`, which estimates the power of a linear model using the F distribution given the effect size of a protein in the discovery study, the degree of freedom based on the replication sample size, and the Bonferroni-corrected significance levels based on the number of proteins selected and available for replication.

The Methods section on this calculation has been revised: "We performed post hoc power calculation to estimate the power for replicating a protein measure based on its effect size from the discovery analysis, the replication sample size, and the Bonferroni-correct significance level given the number of protein measures selected and available for replication using the `pwr.f2.test` function in R. These calculations were conducted for protein measures available for replication from the SomaScan and Olink platforms separately." (page 21, paragraph 2)

2. From lines 581-583, you mentioned that the genetic proxies for protein measure were essentially "cis-pQTLs". I would suggest that you explicitly using this term. This would help clarify the IVs used here, given the three pQTL studies identified both cis and trans pQTLs.

3. Given the authors did not identify the cis-genes with the brain tissue (either MetaBrain or GTEx), I think it is worth discussing that tissue types, i.e. blood and brain, may pinpoint different causal genes/proteins to different neurological diseases/traits. In the introduction section, the authors justified the proteins from blood can be used to study brain diseases, but I think this new discussion/interpretation would be helpful to be added in the manuscript in the Discussion section.

Response: Thank you for this suggestion. Now the manuscript uses *cis*-pQTLs to refer to the genetic proxies of the protein measures used in forward MR.

On the potential difference between blood and brain tissues, the following was added to the Discussion section:

“The cis-pQTLs of these proteins were not significant eQTLs of these proteins in brain tissues suggesting that the protein risk factors for cognitive function may differ between tissues.” (page 15, paragraph 2).

4. For the pQTL datasets used in MR analysis section in line 562, the authors used a meta-analysis result from three studies (Sun et al 2018, Pietzner et al 2021, Ferkingstad et al 2021). Is the “fixed-effect meta-analysis” implemented in METAL or something else? I am asking as I found in lines 519-520, the authors mentioned the tool they used to perform meta-analysis on the differential protein abundance analysis, but not mentioning the tool for the pQTL used for MR. Also, the authors need to provide more details on how they harmonize different human genome reference builds of these three studies given that Sun et al 2018 used hg19, Pietzner et al 2021 used hg19, but Ferkingstad et al 2021 used hg38. Did you liftover hg38 to hg19 before meta-analyzing all three studies?

Response: We indeed used metal for the fix-effect meta-analysis of the pQTLs. About genome build of the 3 pQTL datasets, all summary statistics used rs ID as the identifier, which is stable over time, and using liftover for mapping rs ID across genome build is not recommended (see <https://genome.ucsc.edu/FAQ/FAQreleases.html#snpConversion>). Therefore, the meta-analysis used the rs ID as the common identifier. If rs1 in b37 were merged into rs2 in b38, we would have missed the association results in Ferkingstad et al. for rs1. However, rs2 would be in all 3 datasets. An rs ID in b37 could also be withdrawn from b38. Given that the stability of the rs ID across genome builds and the stringent criteria for selecting genetic proxies (common variants with genome-wide significance and presence in at least 2 pQTL datasets, pruning $r^2 < 0.001$ with 1000G as our pruning reference panel), the impact of the changes in rs IDs from b37 to b38 on our results is likely very small.

The following clarifications have been added in the Methods section: “The meta-analysis was performed using metal with the rs number as the SNP identifier” (page 24 paragraph 1)

Reviewer #3 (Remarks to the Author):

Thank you to the authors for their responses. I have no major outstanding issues. The only comment I would like, given the utility of downstream applications of the results, is to make bulk download of the summary statistics accessible and easy through command line.

Response: The summary statistics were assembled in a zip file and deposited in figshare.